# Synergy of multiple precipitate/matrix interface structures for a heat resistant high-strength Al alloy

Qiang Lu [1], Jianchuan Wang[1], Hongcheng Li[2], Shenbao Jin[2], Gang Sha [2], Jiangbo Lu[3], Li Wang[1], Bo Jin [1], Xinyue Lan[1], Liya Li[1], Kai Li [1,4] ✉ & Yong Du [1] ✉

High strength aluminum alloys are widely used but their strength is reduced as nano-precipitates coarsen rapidly in medium and high temperatures, which greatly limits their application. Single solute segregation layers at precipitate/matrix interfaces are not satisfactory in stabilizing precipitates. Here we obtain multiple interface structures in an Al-Cu-Mg-Ag-Si-Sc alloy including Sc segregation layers, C and L phases as well as a newly discovered χ-AgMg phase, which partially cover the θ′ precipitates. By atomic resolution characterizations and ab initio calculations, such interface structures have been confirmed to synergistically retard coarsening of precipitates. Therefore, the designed alloy shows the good combination of heat resistance and strength among all series of Al alloys, with 97% yield strength retained after thermal exposure, which is as high as 400 MPa. This concept of covering precipitates with multiple interface phases and segregation layers provides an effective strategy for designing other heat resistant materials.

Aluminum alloys are widely used in aerospace, automotive, and other industries due to their outstanding combination of low density, high specific strength, excellent corrosion resistance, and high fracture toughness[1–3]. There exists trade-off between strength and heat resistance of various aluminum alloys. The nano-precipitates with high number densities in high-strength aluminum alloys, such as Al-Cu-based, Al-Zn-Mg-Cu-based, and also in medium-strength Al-Mg-Si-based alloys, will rapidly coarsen during service within the medium and high-temperature ranges, resulting in the deterioration of the alloys' strength[4–7]. On the other hand, aluminum alloys with high heat resistance, such as Al-Mn-based, Al-Si-based alloys, Al-Mg₂Si metal matrix composite contain submicron to micron-scale dispersed phases as the main strengthening phase, leading to their relatively low strength[8,9]. In addition, Al-Sc-Zr-based alloys with Al₃(Sc, Zr) nano-precipitates as the strengthening phase have good thermal stability[10,11], but the yield strength is lower than that of other precipitation-strengthened alloys

such as Al-Cu-based alloys, due to the small volume fraction of the Al₃(Sc, Zr) precipitates.

Solute segregation at grain boundaries or precipitate/matrix interfaces promoted by microalloying elements can effectively improve the strength and/or heat resistance of alloys[12,13]. In many studies on Al-Cu-based alloys, it was found that Sc preferred to segregate at the θ′/Al interface during artificial aging in Al-Cu-Sc alloy and improved the thermal stability of θ′-Al₂Cu precipitates[14–18]. Moreover, Ag was also found to segregate at these precipitate–matrix interfaces in Al-Cu alloys[19,20]. Rosalie and Bourgeois[19] reported that the coherent interface of θ′/Al, i.e., (001)_{θ′} // (001)_{Al}, was decorated by a regular interface structure containing double Ag atomic layers in a high silver content Al-Cu-Ag alloy, which was capable of impeding lateral growth of the θ′ precipitates. In addition, adding trace Ag to Al-Cu-Mg alloy with a high Cu/Mg ratio will promote the precipitation of Ω phase[21–25], which is characterized by the θ-based structure[24] and the AgMg segregation

[1]State Key Laboratory of Powder Metallurgy, Central South University, Changsha 410083, China. [2]School of Material Science and Engineering/Herbert Glitter Institute of Nanoscience, Nanjing University of Science and Technology, Nanjing 210094, China. [3]School of Physics and Information Technology, Shaanxi Normal University, Xi'an 710119, China. [4]Hunan Center for Electron Microscopy, Central South University, Changsha 410083, China. ✉ e-mail: leking@csu.edu.cn; yong-du@csu.edu.cn

layers at the interface with the Al matrix[21,22,26–29]. Hutchinson et al.[26] found that the AgMg segregation layers could hinder the coarsening of the precipitates due to the retarded diffusion of solutes when the thermal exposure temperature was below 200 °C, thereby improving the thermal stability of the alloy. However, when the thermal exposure temperature was higher than 200 °C, the ledges with $1/2\Omega$ unit cell in height were more likely to form at the $\Omega$/Al interface, which would increase the thickening kinetics of $\Omega$ phase. That is, such a single segregation structure at the precipitates/matrix interface still cannot effectively hinder the coarsening of the precipitates.

Co-precipitation of various heat-resistant structures, especially multiple types of heat-resistant precipitates and segregation structures at precipitate/matrix interfaces, is the desire of researchers to pursue simultaneous improvement of strength and heat resistance of Al alloys. Marioara et al.[30] reported that the thermal stability of 6xxx alloys could be improved by reasonably controlling the content of Mg, Si, and Cu to form the fine lath-shaped, Cu-containing, disordered L phase. Therefore, the disordered L phase can be added to Al-Cu-based alloys as a heat-resistant precipitate to improve the heat resistance of Al-Cu-based alloys. The work of Gable et al.[31] and Gariboldi et al.[32] reported that the addition of Si in Al-Cu-Mg-Ag alloys could promote the nucleation of $\theta'$-Al$_2$Cu and the segregation of Mg and Ag solutes at the $\theta'$/Al interface while suppressing the precipitation of $\Omega$. In addition, Si could also promote the formation of C-AlMg$_4$Si$_3$Cu[33] phase and/or L phase (the disordered form of C phase)[32,34], which can act as preferential nucleation sites to promote the heterogeneous nucleation of the $\theta'$-Al$_2$Cu[32,34]. Meanwhile, C-AlMg$_4$Si$_3$Cu phase also can segregate at the $\theta'$/Al interface as an interface phase[32], thereby further improving the heat resistance and strength of Al-Cu-Mg-Ag-Si alloys. The difficulty for co-precipitation of various heat-resistant structures in Al-Cu-Mg-Si-Ag-Sc alloys lies in the following 2 aspects: (1) finding out the appropriate amount of Sc addition to avoid the formation of AlCuSc intermetallic which will consume the Cu solutes; (2) designing the exact concentrations of Mg, Si, and Ag for turning $\Omega$ precipitate into $\theta'$ and C/L phase and promoting the formation of multiple structures such as C/L phase and AgMg layers at the $\theta'$/Al interface.

In this paper, we report an alloy design strategy that can simultaneously provide high strength and heat resistance of aluminum alloys at medium and high-temperature ranges, realized by the synergy of various heat-resistant structures, under the guidance of CALculation of PHAse Diagrams (CALPHAD).

## Results

### The mechanical properties of the designed alloy

An Al-4Cu-0.315Mg-0.5Ag-0.21Si-0.09Sc (in wt%) alloy has been designed by CALPHAD. The calculated phase equilibria at the homogenization/solution temperature of 540 °C and the thermal exposure temperature of 210 °C guided the optimization of solute concentrations step by step, under the principles of: (a) avoiding detrimental AlCuSc phases while introducing Sc for the formation of Sc-rich segregation layers on precipitates, (b) inhibiting the formation of detrimental S-Al$_2$CuMg phase as well as the $\Omega$ precipitate which is not heat resistant enough, and (c) promoting the possible formation of C/L phases and AgMg-rich interface structures at precipitate/matrix interfaces. The CALPHAD details are introduced in the Methods section. As shown in Fig. 1a, compared with the various aluminum alloys thermally exposed at different temperatures for 100 h[4,6,8,35–47], the currently designed alloy shows the best combination of strength and heat resistance after thermal exposure. It reaches a strength retention ratio (the ratio between the yield strength values of an alloy after and before a thermal exposure) of 97% and the highest residual yield strength of 400 ± 5 MPa ever reported for Al alloys thermally exposed at 200 °C for 100 h. This means the designed alloy overcomes the trade-off between heat resistance and strength for different types of aluminum alloys.

According to Fig. 1f, g, the average thickness values of $\theta'$-Al$_2$Cu before and after thermal exposure are 2.8 ± 0.8 nm and 2.9 ± 0.7 nm, while those values for diameter are 43 ± 20 nm and 46 ± 20 nm, respectively. The statistical results show that the main strengthening phase $\theta'$-Al$_2$Cu has not been obviously coarsened after thermal exposure, explaining the alloy's high strength retention rate shown in Fig. 1a.

### Multiple types of precipitates and interface structures in peak-aged state

According to the atom probe tomography (APT) results shown in Supplementary Fig. 1 and atomic resolution Z-contrast high-angle angular dark field-scanning transmission electron microscopy (HAADF-STEM) images shown in Fig. 2, the main precipitates are plate-like $\theta'$-Al$_2$Cu on {001}$_{Al}$ planes, intergrowing with them are some plate-like C precipitates on {001}$_{Al}$ planes and lath-like L precipitates along <001>$_{Al}$ directions. Figure 2b–e displays the structures of the plate-like $\theta'$-Al$_2$Cu precipitates, surrounded by C and L interface phases, while Fig. 2f shows one of the independently precipitated L laths. The L phase is the disordered form of the C-AlMg$_4$Si$_3$Cu[33] phase (space group P2$_1$/m, see Supplementary Data 1). L was reported with high thermal stability in Al-Mg-Si-Cu alloys[48,49]. In addition to C or L precipitates, there is a type of interface phase containing 3 atomic layers of Ag and Mg solutes at $\theta'$/Al interfaces, as shown in Fig. 2c, e, g. This new type of interface phase is named $\chi$.

The existence of the interface phases changed the interface structures of $\theta'$-Al$_2$Cu precipitates and was found to hinder their coarsening. Bourgeois et al.[50–53] reported that Cu atoms could occupy the interstitial sites at the coherent interface of $\theta'$-Al$_2$Cu and reduce the energy of the system. It is concluded that the Cu interfacial segregation layer is an intermediate state beneficial for the thickening of $\theta'$-Al$_2$Cu. In this study, the intensity line profile of A1-A2 in Fig. 2b–e indicates that, at the coherent interfaces without AgMg or C interface phase, Cu atoms can occupy the interstitial sites at the interfacial layer. In Fig. 2c, it is found that $\chi$-AgMg interface phase (on the left side) effectively hinders the thickening of the $\theta'$-Al$_2$Cu precipitates, while in the right part with only layers containing interstitial Cu atoms there is abrupt thickening (see the red arrow in the figure). Figure 2g also shows the relationship between the interface phases and $\theta'$-Al$_2$Cu, demonstrating that the $\chi$-AgMg (in green color) and plate-like C (in purple color) interface phases mainly exist at the coherent $\theta'$/Al interface, while there is only disordered L phase (also in purple color) at the semi-coherent $\theta'$/Al interface. In addition, there are segregation layers of Sc solutes at the $\theta'$/Al interfaces according to Fig. 2h, i, which can further retard the diffusion of Cu elements. In comparison, an Al-4Cu control alloy has also been prepared in the same way as for the current Al-4Cu-0.315Mg-0.5Ag-0.21Si-0.09Sc alloy. As can be seen in Supplementary Figs. 2, 3, the unsatisfactory thermal stability of the peak-aged Al-4Cu alloy intuitively reveals the crucial role the multiple interface structures play in effectively hindering the coarsening of $\theta'$-Al$_2$Cu precipitates in the current alloy.

### The structure of the newly discovered $\chi$-AgMg interface phase

The structure of the newly discovered $\chi$-AgMg interface phase found in Fig. 2 is different from the uniformly distributed AgMg bi-layer at the $\Omega$/Al interface reported by Kang et al.[21], or the Ag segregation layer with double atomic layers at the $\theta'$/Al interface reported by Rosalie et al.[19]. According to Fig. 3a–d, the atomic arrangement of the interface phase is consistent with that of Al under different zone axes, with the light and dark atomic columns alternately arranged at sub-layers L1 and L3. These atomic columns were identified as Ag and Mg columns according to the atomic resolution energy dispersive X-ray (EDX) mapping results shown in Fig. 3e–h. As for the middle layer L2, the EDX maps show that every column contains Ag, while there is a very low signal of Mg. However, the intensity line profiles of the middle layer L2, as inserted in Fig. 3a, c, show that the column intensity is in an alternative strong-weak-strong

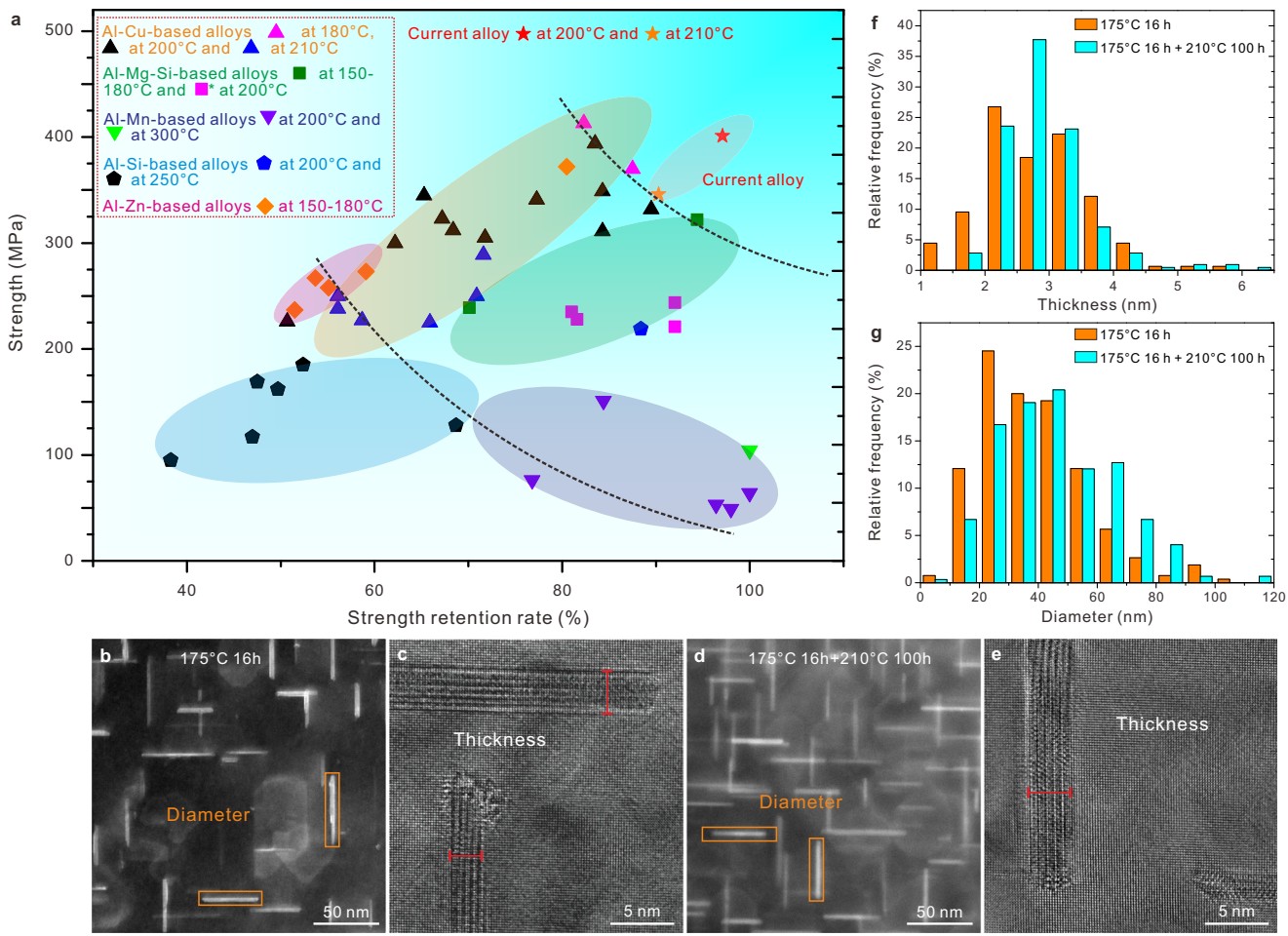

**Fig. 1 | Mechanical properties and size stability of precipitates of the current Al-4Cu-0.315Mg-0.5Ag-0.21Si-0.09Sc alloy. a** Yield strength (superscript* for tensile strength) of different Al alloys thermally exposed at different temperatures for 100 h[4,6,8,35–47] and corresponding strength retention rates, more details can be found in Supplementary Table 1. **b, c** Typical images used to measure the diameter and thickness of θ′-Al₂Cu precipitates in the peak-aged state of the current alloy. **d, e** Typical images for the state after thermal exposure. **f, g** The distributions of thickness and diameter of the current alloy, respectively, for both the peak-aged and thermally exposed states.

distribution, implying the atomic column with lower intensity should also contain Mg. As shown in Fig. 3f, Mg signals can be found at the atomic columns marked by white dotted circles in the L2 layer. Similar examples can be found in Supplementary Fig. 4. In addition, the atomic ratio of Mg in the three layers is about 2:1:2 according to concentration line profile of Mg shown in Fig. 3i, further confirming that the L2 layer contains Mg atoms. According to the information mentioned above, the structure of χ-AgMg interface phase was constructed as shown in Fig. 3j, while the atomic ratio of Ag to Mg in the structure is 1.4. Supplementary Figs. 5, 6 show the compositions of χ-AgMg interface phase detected by EDX and APT, with Ag/Mg ratios of 1.56 and 1.44, respectively. These ratios are all close to the atomic ratio of Ag to Mg in the constructed structure. Furthermore, the images simulated by QSTEM software[54] along [100], [010], and [110] directions of the constructed structure are all consistent with the experimental images. Meanwhile, to visually compare the newly discovered χ-AgMg interface phase with the AgMg bi-layer at the Ω/Al interface[21] reported by Kang et al. and the Ag segregation layer at the θ′/Al interface reported by Rosalie et al.[19], the structure of these three AgMg or Ag segregation structures were shown in Supplementary Fig. 7, while the crystallographic information of these three structures was also uploaded as Supplementary Data 2–4.

**Nano-scale precipitates after thermal exposure**

According to Fig. 4a–c, the θ′-Al₂Cu, L, and C phases in the current alloy were kept after a thermal exposure at 210 °C for 100 h. However,

there are some square σ-Al₅Cu₆Mg₂ (space group Pm-3)[55] particles in the thermally exposed sample as shown in Supplementary Fig. 8. In addition, according to Figs. 1d, 4d, the thin χ-AgMg interface phase in the peak-aged state disappeared after thermal exposure, while a thicker AgMgAl phase, which was discovered in this work and named ξ phase, precipitated at the σ/Al interface. Figure 4d, e shows the atomic resolution HAADF-STEM images and the corresponding Fast Fourier Transform (FFT) pattern, respectively. As shown in the FFT pattern, the distance $g_1$ is 5.17 nm$^{-1}$, while those of $g_2$ and $g_3$ are 4.20 and 2.99 nm$^{-1}$, respectively. That is, the ratios $g_1/g_3$ and $g_2/g_3$ are 1.729 and 1.404, which are all consistent with the standard electron diffraction pattern of the body-centered cubic (BCC) structure along the [011] direction. In addition, the intensity line profile inserted in Fig. 4d shows that the light and dark atomic columns alternately arranged along A1 to A2, which could be identified as Ag and Mg+Al atomic columns according to the EDX mapping results shown in Fig. 4f. It is obvious that Mg and Al atoms jointly take one site in the unit cell, most probably in a disordered way, while Ag atoms solely occupy the other site. According to the information mentioned above, the B2 structure of ξ-Ag₁Mg₁₋ₓAlₓ ($x = 0.5$) phase has been constructed as inserted in Fig. 4d. The space group of the ξ phase is Pm-3m, while the lattice parameters are determined as $a = b = c = 3.34 \pm 0.10$ Å and $\alpha = \beta = \gamma = 90°$. The atomic ratio Ag: Mg: Al in ξ phase is 2:1:1, which is close to the ratio of about 51: 22: 28 as obtained from the EDX data (from the very thin area shown in Fig. 4d). Furthermore, the HAADF-STEM image (see the blue frame in

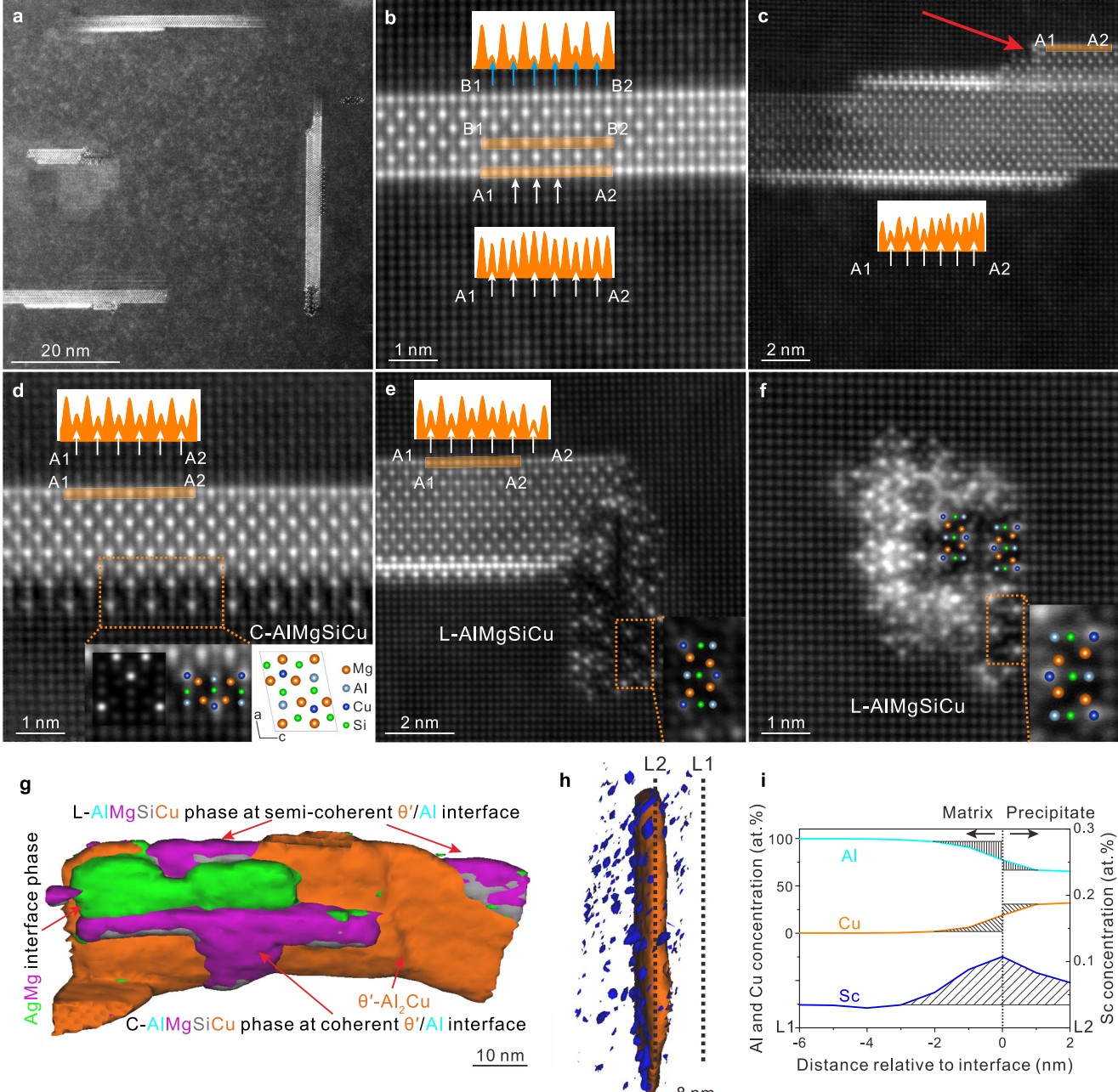

**Fig. 2 | Multiple types of precipitates and interface structures of the current alloy under peak ageing. a** Low magnification HAADF-STEM image. **b** Individual θ'-Al₂Cu. **c** AgMg interface phase occurring at both upper and lower θ'/Al interfaces. **d** θ'-Al₂Cu with C interface phase at the coherent interface. **e** θ'-Al₂Cu with a L phase at the semi-coherent interface. **f** Independently precipitated L. The unit cell structure of the C phase and the simulated HAADF-STEM image by QSTEM have been inserted in (**d**). The intensity line profiles inserted in the images show the intensity variation of Cu columns in different interface layers. **g, h** Multiple interface phases and segregation layers at the θ'/Al interfaces detected by APT. **i** Proxigram of Sc cross the θ'/Al interface along L1 to L2 in (**h**).

Fig. 4d) simulated by QSTEM software[54] along [011] direction of ξ phase is consistent with the experimental image. The crystallographic information file is uploaded as Supplementary Data 5.

### The under-aged microstructure

In the currently designed alloy, the main precipitates are the θ'-Al₂Cu and C/L phases during artificial ageing treatment. The fact that C/L precipitates act as heterogeneous nucleation sites to promote the formation of θ'-Al₂Cu is confirmed by the intergrowing behaviors of C/L and θ' in the under-aged alloy according to the APT results shown in Fig. 5a–c, Supplementary Fig. 9 and Supplementary Movie 1. The main precipitates in this state are laths and tiny plates enriched with Mg, Si,

and Cu, which correspond to L and C phases, while θ' has just nucleated with a much smaller size. The formation of C/L precipitates in the early stage of ageing consumed a substantial fraction of Mg solutes, thus there were insufficient MgAg atomic clusters to form the Ω during the subsequential stages of ageing. As a result, θ' phase was precipitated instead. Then, the remaining Mg solutes combined with Ag solutes to form the newly discovered χ-AgMg interface phase that partially covers the θ' precipitate.

### Discussion

The precipitation driving force of different phases is used to explain the precipitation sequence in the current alloy during the artificial

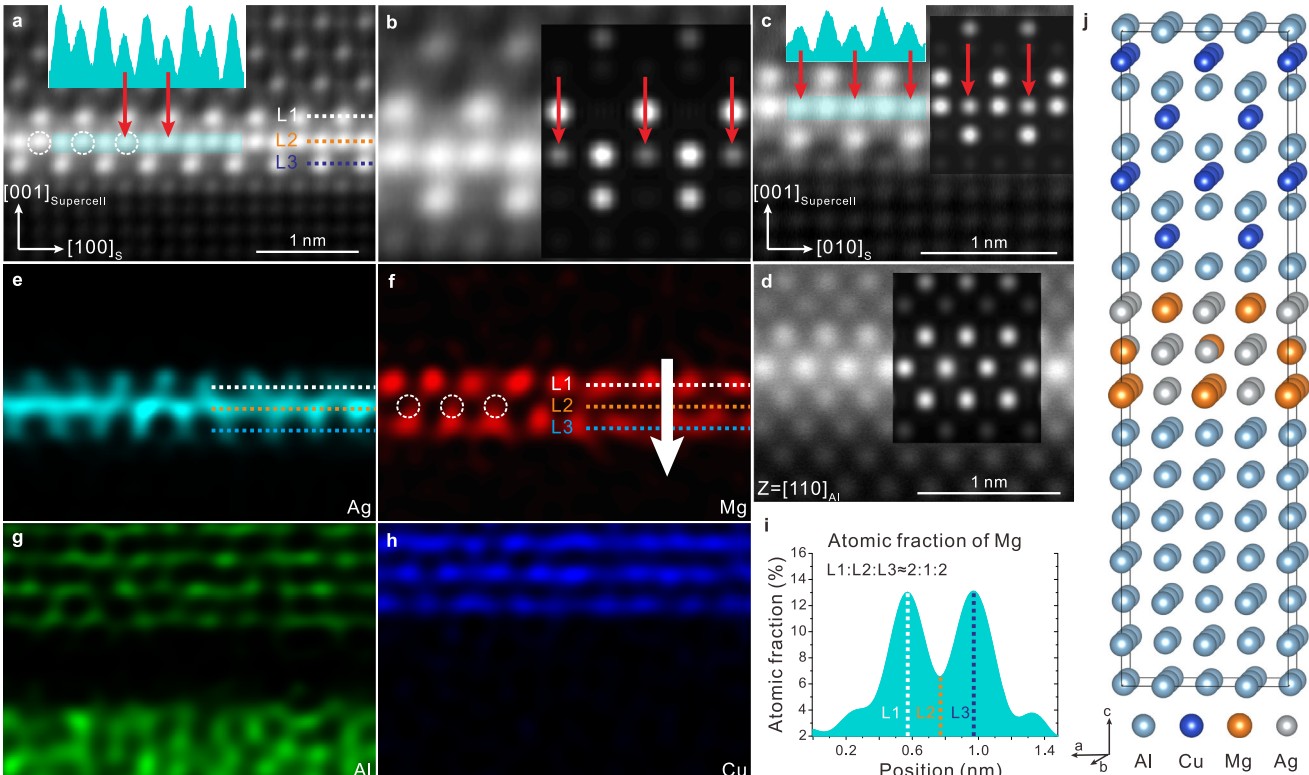

**Fig. 3 | The structure of the newly discovered χ-AgMg interface phase. a–c, d** Atomic resolution HAADF-STEM images of the χ-AgMg phase along [100]$_{Al}$ and [110]$_{Al}$, respectively. **b** Enlargement image of (**a**). **e-h** Atomic resolution EDX elemental maps of the area shown in (**a**). The intensity line profiles of the middle layer (L2) of the χ-AgMg phase are inserted in (**b**) and (**c**). **i** Concentration line profile of Mg in the χ-AgMg phase along the white arrow, the integration width is 7 nm. **j** 3D

model of the θ'/Al interface structure with the χ-AgMg phase. It should be noted the viewing direction is parallel to [010] direction of the supercell in (**j**) for (**a**, **b**), [100] for (**c**), and [110] for (**d**). The HAADF-STEM images simulated by QSTEM using the constructed model along different directions are inserted in (**a**), (**b**), and (**c**). The atomic columns marked with white dotted circles in (**f**) correspond to those similarly marked in (**a**).

ageing heat treatment. Due to the lack of thermodynamic model of metastable C/L phase in the newest version of multi-component multi-phase thermodynamic database TCAL for Al alloys[56], the precipitation driving forces of successors of C/L and θ', i.e., the equilibrium Q-AlMgSiCu and θ-Al$_2$Cu phases, respectively, were calculated instead. As shown in Fig. 5d, the precipitation driving force of Q phase increases rapidly when Si is added to the Al-Cu-Mg-Ag alloy, while that of θ phase remains almost unchanged. The precipitation driving forces of Q and θ in the currently designed alloy are 11.69 kJ mol$^{-1}$ and 3.227 kJ mol$^{-1}$ (mol for atoms), respectively, which imply the C/L phases, as precursors of Q phase, will preferentially precipitate.

Ag solutes not only participate in the formation of the χ-AgMg interface phases at the θ'/Al interface, but also segregate in C/L precipitates. This is revealed by the co-segregation of Mg, Si, Cu, and Ag solutes in the APT results in Supplementary Figs. 1, 9 and the elements concentration profiles in Fig. 5b, c. Weng et al.[57] reported that Ag could promote the nucleation of the precipitates and increase their number density in Al-Mg-Si-Ag alloys, thereby improving the mechanical properties of the alloys. As for the current Al-Cu-Mg-Ag-Si-Sc alloy, Ag solutes probably also played an important role in increasing the number density of C/L phase, thereby increasing the number density of θ'. This accounts for the high strength of the alloy. In addition, as typically shown in Fig. 2f, 4c and Supplementary Fig. 10, Ag was found to enrich at the interface of L precipitates to form a segregation layer, thereby further increasing the thermal stability of the disordered L phase.

To compare the effects of the various interface structures on the stability of the precipitates more accurately, the formation energies of different structures were calculated, as shown in Fig. 6. The formation energy of Model 3 is lower than that of Model 2, indicating that the Al-

terminated θ'-Al$_2$Cu interface structure (with the χ-AgMg interface phase) is easier to form and more stable than Cu-terminated θ'-Al$_2$Cu interface structure. In addition, Model 1 shows a single χ-AgMg interface phase embedded in the matrix, with a higher formation energy than those of Model 2 and Model 3. That is, the χ-AgMg interface phases cannot precipitate alone and should exist in the coherent interface of θ'/Al. Therefore, the most energetically favorable structure containing χ-AgMg interface phase is Model 3, which is consistent with the experimental results shown in Fig. 3. In addition, the formation energy of Model 6 is obviously lower than those of Models 4, 5, indicating that the C interface phase layer can improve the stability of θ'-Al$_2$Cu more efficiently than the interstitial Cu layer. Especially, the θ'/Al interface model containing a C interface phase is most energetically favored among all the six models.

As for other θ'/Al interfacial positions without interface phases, the stability could be enhanced by the covering of Sc solutes. According to calculations by Bourgeois et al.[19], when the thickness of a θ' precipitate is less than 4 nm, the Cu atoms tend to completely occupy the interstitial sites in the Cu-terminated interface. According to Fig. 1f, most θ' plates were found to be thinner than 4 nm in the peak-aged state of the current alloy. Therefore, interstitial Cu layers were found at most coherent θ'/Al interfaces in our alloy, in addition to χ-AgMg and C interface phases. Moreover, according to Fig. 2h, i, Sc segregation layers were also found at θ'/Al interfaces. Zhang et al.[58] investigated the effect of the Sc segregation layer on the stability of interface of θ'/Al by first-principles calculations. Their results showed when Sc segregated at the interface of θ'/Al, the bonding at the interface containing the Cu interstitial layer can be greatly enhanced relative to that without a Cu interstitial layer. Such strong bonding is beneficial for inhibiting the thickening of θ'. These calculations explain

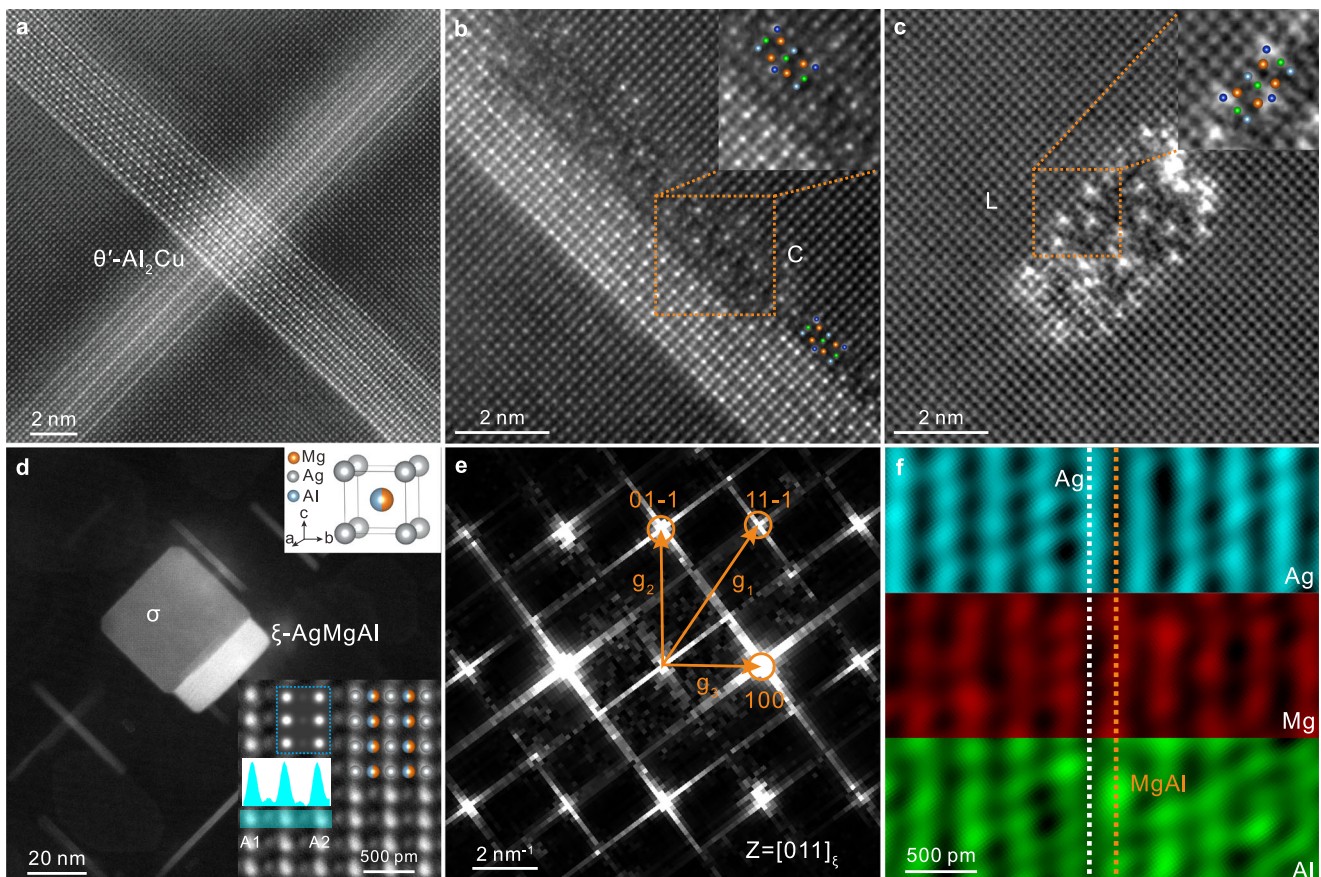

**Fig. 4 | HAADF-STEM images and EDX results of precipitates after thermal exposure at 210 °C for 100 h. a** HAADF-STEM image of θ′-Al₂Cu. **b** C interface phase at the θ′/Al interface. **c** Independently precipitated L phase. **d** Low-magnification and atomic resolution HAADF-STEM images of the ξ phase.

**e** Corresponding FFT pattern of atomic resolution HAADF-STEM image in (**d**). **f** EDX elemental maps of ξ phase. The unit cell of the ξ phase and the simulated HAADF-STEM image along (011)ξ by QSTEM have been inserted in (**d**). The intensity line profiles inserted in (**d**) shows the intensity variation from A1 to A2.

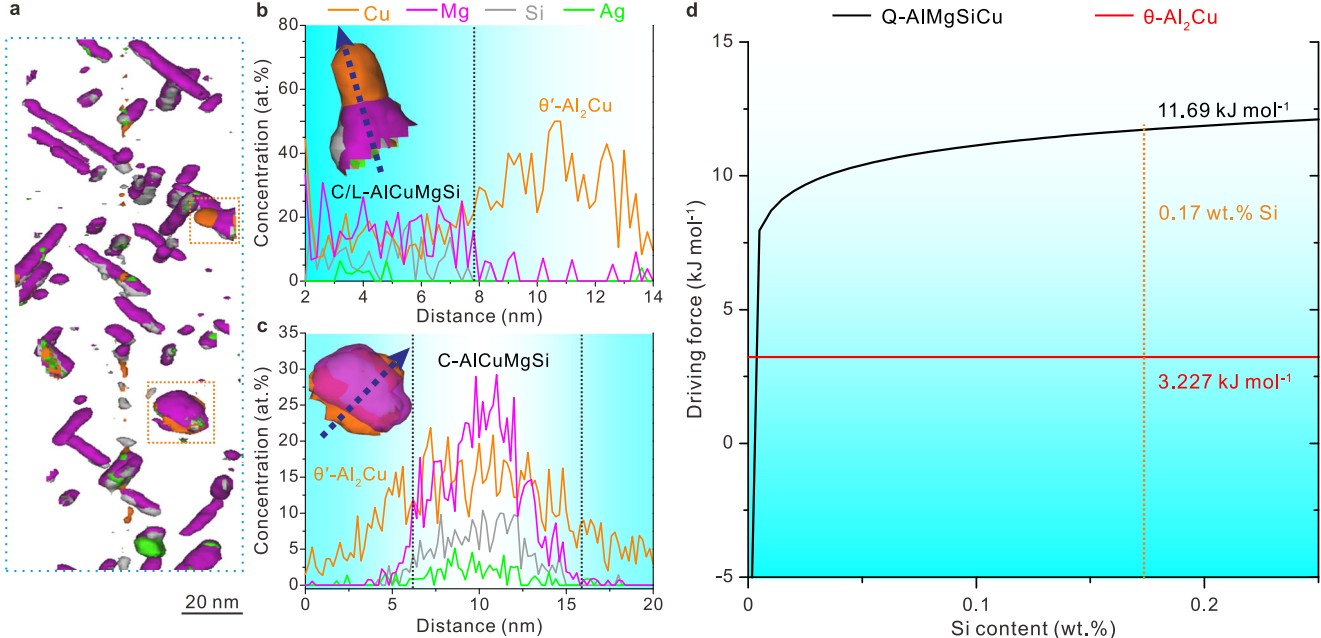

**Fig. 5 | The APT results of precipitates and calculated driving force. a** APT result of the current alloy in the under-aged state (at 175 °C for 1 h). **b**, **c** Concentration profiles of the precipitates along the blue arrows. **d** Driving force of equilibrium Q-AlMgSiCu and θ-Al₂Cu at the ageing temperature of 175 °C.

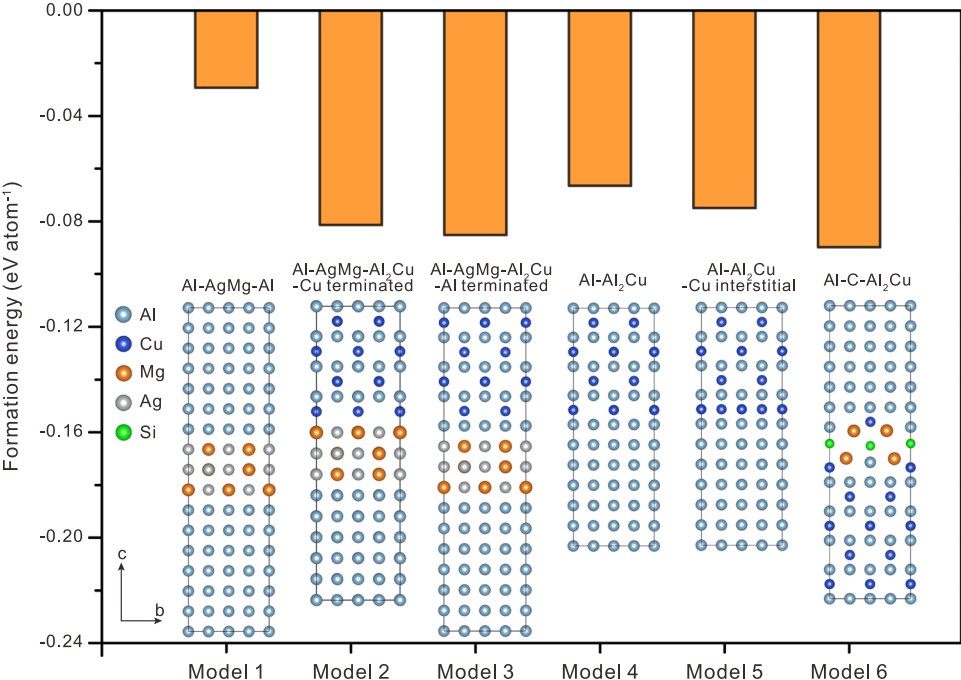

**Fig. 6 | The formation energies of different interface structures.** Model 1 shows the structure in which the χ-AgMg phase exists alone in the Al, while Models 2 and 3 are the structures in which the χ-AgMg phase exists in the θ′/Al interface with Cu terminated and Al terminated, respectively. Models 4 and 5 represent the θ′/Al interface structure without and with Cu interstitial atoms, respectively. Model 6 is the structure in which the C phase exists in the θ′/Al interface.

the high stability of θ′ precipitates partially covered with Sc segregation layers observed in our work and many other studies[14,16,18].

In addition to interface structures at the coherent interface of θ′, there are always disordered L precipitates at the semi-coherent interface of θ′, i.e., occurring at one end of a θ′ plate (see Fig. 2e and Supplementary Movie 2). In the early stage of artificial ageing process, the disordered L phase can serve as heterogeneous nucleation sites to promote the nucleation of the θ′[34], thereby reducing its size and increasing the number density[30]. On the other hand, broadening of θ ′-Al$_2$Cu precipitates will also lead to an obvious decrease in the number density of θ′-Al$_2$Cu precipitates, leading to a decrease in the alloy properties according to the work of Nie et al.[59]. In the current alloy, due to the high thermal stability of disordered L phase[30,48], the broadening of θ′ at semi-coherent interface can be hindered during the long-time thermal exposure process. Therefore, the integration of multiple types of interface structures including χ-AgMg interface phase, C interface phase, and Sc segregation layers at coherent interface and the L phase at semi-coherent interface could stabilize the interface of θ′-Al$_2$Cu and suppress the coarsening of the θ′-Al$_2$Cu precipitates, thereby improving the heat resistance of the current alloy.

Ab initio molecular dynamics (AIMD) simulations has been performed to account for the stability of χ-AgMg and C interface phases during thermal exposure. Experimentally, the C interface phase appeared more stable than the χ-AgMg interface phase during thermal exposure. As shown in Fig. 7a, The AIMD results calculated at 1073 K reveal that Ag atoms close to the matrix gradually cross the interface and enter the aluminum matrix, while Al atoms in the interface diffuse into the χ-AgMg interface phase in turn. Therefore, the χ-AgMg interface phase gradually dissolved during thermal exposure. On the contrary, the C interface phase in Fig. 7b shows high stability at high temperatures and the structure can hardly be destroyed, which is consistent with the experimental results shown in Fig. 4b. Therefore, the C interface phase can more effectively hinder the growth of θ′ precipitates than the χ-AgMg interface phase during long-term thermal exposure.

The high number density as well as uniform size distribution of precipitates in the current Al-4Cu-0.315Mg-0.5Ag-0.21Si-0.09Sc alloy,

as shown in Fig. 5a and Supplementary Fig. 1, probably acts as one of the factors in hindering the coarsening of the precipitates. This has been validated in the work of Makineni et al.[60] where Nb and Zr additions increased the number density of θ′ and the heat resistance of their Al-2Cu-0.1Nb-0.15Zr alloy. However, in our work, by comparing Supplementary Fig. 3 with Fig. 1, it can be found the θ′ precipitates are even finer, and the number density is probably significantly higher in the Al-4Cu control alloy processed in the same way as the current alloy, while its heat resistance is much worse. Thus, it is suggested the presence of the multiple interface structures at the θ′/Al interface is the main cause of effectively inhibiting the coarsening of θ′ precipitates in the current alloy.

In conclusion, a composite nanostructure containing C interface phases, Sc-rich segregation layers, and a newly discovered χ-AgMg interface phase at the coherent interface of θ′ precipitates, L phase at semi-coherent interfaces, as well as independently precipitated L phase, has been obtained in an Al-Cu-Mg-Ag-Si-Sc alloy. These interface structures effectively retarded the coarsening of θ′. Thus, the strength and heat resistance of the alloy were simultaneously improved. In addition, as the current alloy was fabricated by traditional processing technologies, it exhibits great potential in industrial application. Meanwhile, this design concept of co-segregation/precipitation at precipitate/matrix interfaces, by reasonably controlling the content of each element using CALPHAD approach, can provide a reference for the design of other heat-resistant materials.

## Methods
### Alloy design by CALPHAD
The contents of Cu, Mg, Si, Ag, and Sc were reasonably determined by CALPHAD. All the thermodynamic calculations were performed in Thermo-Calc (TC) software based on the multicomponent multiphase thermodynamic database for Al alloys (TCAL5)[56,61]. The design principles are as follows: (1) the formation of AlCuSc phase during the homogenization should be avoided to ensure that the main alloying elements such as Cu and Sc are mostly dissolved into the matrix; (2) the generation of harmful precipitates such as S-Al$_2$CuMg at high

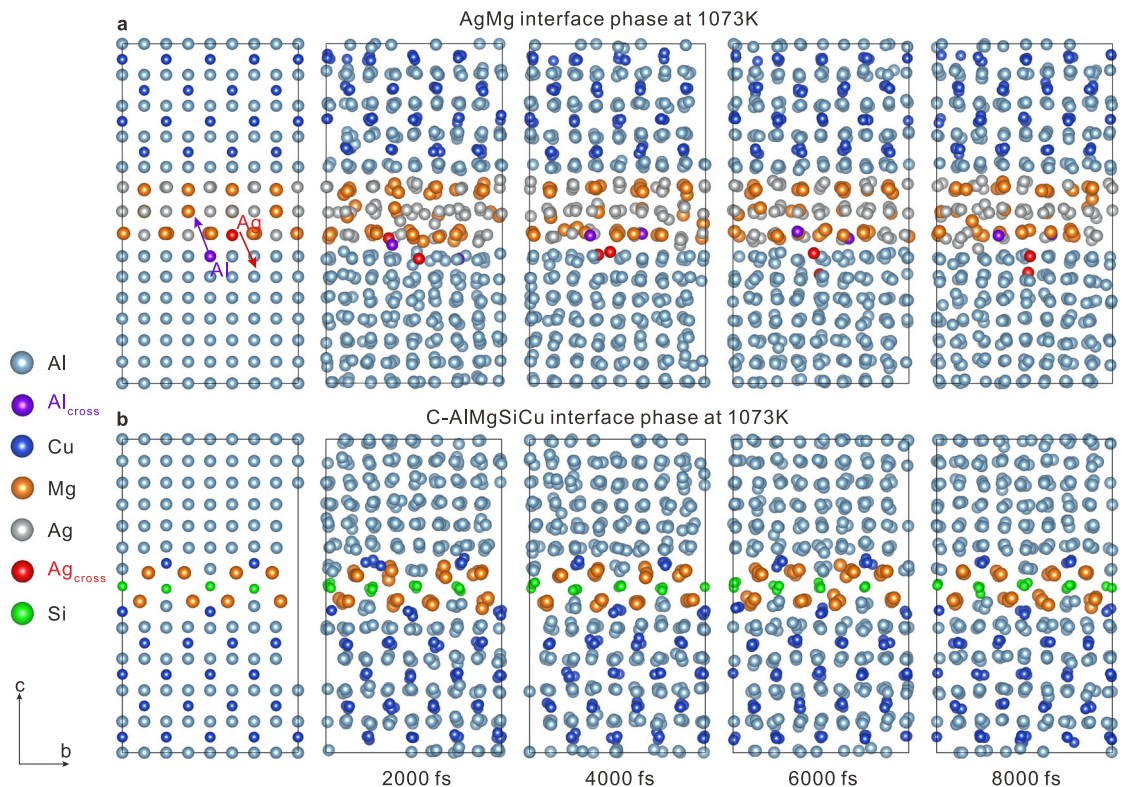

**Fig. 7 | The AIMD results of different interface phases calculated at 1073 K. a** AIMD results of χ-AgMg interface phase. **b** AIMD results of C-AlMgSiCu interface phase. The Ag and Al atoms crossing the layers are marked in red and purple, respectively.

temperatures should be inhibited during thermal exposure, meanwhile the Ω precipitate which is not heat resistant enough should be suppressed. (3) the formation of multiple types of structures such as C/L interface phase, AgMg-rich, and Sc-rich segregation layers at the interface of θ′/Al is attempted. The specific calculation process of alloy design by CALPHAD is described as follows:

**Control of Cu content.** As shown in Supplementary Fig. 11a, when Cu content of an Al-xCu alloy is set as 5 wt%, the liquid phase is generated when the temperature is above 557 °C while the equilibrium phase θ-$Al_2Cu$ will be produced when the temperature is below 532 °C. Therefore, the homogenization temperature should be selected between 532 and 557 °C to ensure that Cu can be completely dissolved into the Al matrix. However, due to the narrow temperature range, it is easy to cause over-burning or insufficient homogenization due to the temperature deviations and fluctuations during the homogenization process. Therefore, it is necessary to reduce the Cu content to obtain a wider homogenization temperature range. When Cu content is 4 wt%, the homogenization temperature could be selected from 507 °C to 571 °C, which is wider than that for Al-5Cu alloy. Hence, the Cu content is set as 4 wt%. The specific homogenization temperature is set as 540 °C, close to the average value of the temperature range.

**Control of Sc and Ag contents.** As the homogenization temperature is 540 °C, the ternary isothermal section of Al-Cu-Sc system is calculated at 540 °C as shown in Supplementary Fig. 11b. When the content of Cu is 4 wt%, the maximum solid solubility of Sc element in the Al-Cu-Sc alloy is 0.09 wt%. To suppress the formation of AlCuSc which will consume Cu during the homogenization heat treatment, the content of Sc in the currently designed alloy is set as 0.09 wt%. In this condition, the formation of AlCuSc has been effectively suppressed as shown in Supplementary Fig. 12. When it comes to the Ag content, although it has a high solid solubility in the aluminum matrix at 540 °C, its content should be controlled at the thermal exposure temperature

210 °C to avoid the formation of equilibrium phase named HCP_AlAg. According to Supplementary Fig. 11c, HCP_AlAg will be generated when the content of Ag is above 0.51 wt% in the case of 4 wt% Cu. Therefore, the content of Ag in the currently designed alloy is set as 0.5 wt%.

**Control of Mg and Si contents.** In order to obtain θ′-$Al_2Cu$ precipitates with Sc and AgMg segregation layers in Al-Cu-Mg-Ag-Si-Sc alloy, it is necessary to control the Mg/Si ratio to promote the precipitation of θ′-$Al_2Cu$ while inhibiting the formation of Ω. Gable et al.[31] reported that when the Mg/Si mass ratio is below 2, the Ω precipitate in the Al-Cu-Mg-Ag alloy will be completely suppressed. Therefore, the composition should be below the red dotted line representing the Mg/Si mass ratio of 2 in Supplementary Fig. 11d. In addition, the content of Mg should be higher than that of Si (i.e., above the phase boundary indicated by the black arrow) to ensure that there is no silicon particle. Finally, the Mg/Si mass ratio in the currently designed alloy is set as 1.5, as shown by the yellow dotted line in Supplementary Fig. 11d, which is in the middle of the red dotted line and the black solid line. Meanwhile, the S-$Al_2CuMg$ phase will not be generated in this combination of temperature and composition. It should be noted that the phase equilibria of Al-Mg-Si-4Cu (in wt%) at 175 °C (ageing temperature) are very similar to those at 210 °C (thermal exposure temperature).

The Mg content in the currently designed alloy will be divided into two parts. One part is used to form the C-$AlMg_4Si_3Cu$ or L phase (disordered form of C phase), which is the main heat-resistant precipitate. The other part of Mg will participate in the formation of the AgMg segregation layer at the θ′/Al interface. However, there is currently no research report on the composition of the AgMg layer at the θ′/Al interface, although it was indeed found in some conditions[32]. Therefore, the Mg/Ag atomic ratio in the AgMg layer segregated at the θ′/Al interface is based on the ratio reported for the AgMg segregation layer at the Ω/Al interface. Kang et al.[21] constructed a AgMg bi-layer model at the Ω/Al interface, in which the Ag/Mg atomic ratio is about 2:1. However, the APT results of Reich et al.[22] and Bai et al.[62,63] show that

the Ag/Mg atomic ratio in the AgMg layer is about 1:1 in the peak-aged and over-aged states. Therefore, the Ag/Mg atomic ratio is set as 1.5 in this paper, which is the intermediate value between theoretical predictions and experimental results. The AgMg segregation layers were expected, although during the design it was not known what their atomistic structures would be. According to the ratio of each element as mentioned above, the contents (in wt%) of Mg and Si, i.e., $W(Mg)$ and $W(Si)$, can be calculated according to the following Eqs. (1–3):

$$\frac{W(Mg)_C}{Ar(Mg)} = \frac{4}{3} * \frac{W(Si)}{Ar(Si)} \tag{1}$$

$$\frac{W(Mg)_{AgMg}}{Ar(Mg)} = \frac{2}{3} * \frac{W(Ag)}{Ar(Ag)} \tag{2}$$

$$W(Mg)_C + W(Mg)_{AgMg} = \frac{3}{2} * W(Si) \tag{3}$$

in which $W(Mg)_C$ is the part of Mg used to form the C-AlMg$_4$Si$_3$Cu phase, while $W(Mg)_{AgMg}$ is the other part used to form the AgMg segregation layer at the θ′/Al interface. The relative atomic masses of Mg, Si, and Ag are represented by $Ar(Mg)$, $Ar(Si)$, and $Ar(Ag)$, respectively. After calculations, the contents of Si and Mg are set as 0.21 wt% and 0.315 wt%, respectively. Therefore, the final alloy composition is determined as Al–4Cu–0.315Mg–0.5Ag–0.21Si–0.09Sc (in wt%).

## Sample preparation

The designed alloy was melted under argon atmosphere and cast into an iron mold with a size of Φ20 mm × 10 mm. The as-cast ingots were homogenized at 540 °C for 16 h and subsequently hot- and cold-rolled into 1.5-mm-thick sheets. After homogenization, the composition of alloys was detected by Spectro Blue SOP inductively coupled plasma optical emission spectrometer. Solution heat treatment of the sheet samples was conducted at 540 °C for 60 min, followed by water quenching. The quenched samples were then artificially aged at 175 °C for 16 h to reach the peak-aged state. Furthermore, peak-aged samples were subjected to thermal exposure heat treatment for 100 h at 210 °C or 200 °C. The former temperature was selected to test the heat resistance in a higher temperature than that usually used, while the latter temperature was selected for comparison with available literature data.

## Mechanical performance tests

An Instron 3369 mechanical testing machine was used to perform the tensile tests at a constant speed of 2 mm min⁻¹. The tensile data is the average value of three or two parallel tensile samples in the same condition. The Microhardness of the various aged Al-4Cu samples was tested by HV-1000IS automatic turret Vickers microhardness tester, applying a load of 100 g and a dwell time of 15 s. To ensure the reliability of the data, ten tests were performed on each sample, and the microhardness is obtained by averaging the data points.

## Nano-to-atomic scale characterizations

TEM samples were electron-polished by a Struers TenuPol-5 twin-jet electro-polishing instrument using a solution of 70 vol.% methanol and 30 vol% nitric acid at −30 °C. An FEI Titan G2 60-300 TEM instrument with an objective spherical aberration corrector operated at 300 kV was used to observe the size change of precipitates before and after thermal exposure treatment. The atomic resolution HAADF images and EDX elemental maps of peak aged alloy were taken on a Thermo Fisher Titan cubed Themis G2 300 TEM instrument operated at 80 kV with a probe spherical aberration corrector and a super-X high-resolution EDX system, while the atomic resolution HAADF images and EDX elemental maps of the alloy after thermal exposure were taken on a Thermo Fisher Spectra 300 TEM instrument operated at 300 kV with a probe spherical

aberration corrector and a super-X high-resolution EDX system. Each atomic resolution HAADF image of peak aged alloy was acquired by superimposing 50 fast scan images recorded with the drift-corrected frame integration technology. In addition, to obtain high enough signals for the atomic resolution EDX maps of the peak aged alloy detected at 80 kV, the total acquisition time was about 26 min.

TEM data analysis was conducted on the software package Velox version 3.0.0.815. The atomic resolution HAADF images after thermal exposure were filtered using Gaussian blur or Radial Wiener, while all the atomic resolution EDX images were filtered using Radial Wiener in Velox software. The atomic arrangement and the intensity variations of the atomic columns in the filtered image are consistent with the original image. Referring to the method proposed by Wenner et al.[64] for calculating the signal-to-noise ratio (SNR), the residuals of the Gaussian fit are used as the noise and the fit height as the signal. The SNR of EDX elemental maps shown in Fig. 3 is higher than 3, which indicates that the results of EDX maps are reliable. In addition, the relative standard deviation of the content of different χ-AgMg interface phases detected by TEM is below ±3%. Furthermore, the Mg concentration line profile of the χ-AgMg interface phase shown in Fig. 3i is obtained with a 7 nm integration width perpendicular to the line, by applying the Intensity Profile tool in the Velox software to the quantified elemental-mapping data, which have been background corrected and peak fitted.

The samples for APT analysis were thin bars of 0.5 × 0.5 × 20 mm³ cut from the sheets and further thinned into fine needles by the standard two-step electro-polishing procedure. The APT needle samples were tested in a LEAP 4000 HR instrument at a temperature of 20 K and a pulse repetition rate of 200 kHz. Data reconstruction and analysis were conducted on the software package IVAS version 3.8.2.

## First-principles calculations

All first-principles calculations have been performed within the framework of density functional theory (DFT), as implemented in the Vienna ab initio Simulation Package code[65]. Projector-augmented wave potentials[66] and Perdew–Burke–Ernzerhof within generalized gradient approximation[67] were used to treat the ion-electron interaction and electron exchange-correlation effects, respectively. The cut-off energy was set to be 500 eV for plane-wave expansion for wave function. Models containing θ′ and C phases were constructed based on the experimental matrix/precipitate orientation relationship $(0\ 0\ 1)_{θ′}$ // $(0\ 0\ 1)_{Al}$, $[1\ 0\ 0]_{θ′}$ // $[1\ 0\ 0]_{Al}$; $(010)_C$ // $(0\ 0\ 1)_{Al}$, $(001)_C$ // $(1\ 0\ 0)_{Al}$[53]. The dimensions of each model parallel to the interfacial plane (cell vectors **a** and **b**) were expanded, resulting in a 2 × 2 geometry with dimensions | **a** | = 2a$_{Al}$ and |**b**| = 2a$_{Al}$. In this condition, the mismatch of θ′ with aluminum matrix is 1.18%, indicating the constructed interface is reliable. The different structure models were shown in Fig. 6. The Brillouin zone was sampled with Gamma-center scheme using the **k**-points grid of 17 × 17 × 17 for bulk, 3 × 3 × 1 and 5 × 5 × 1 for relaxation calculation and total energy calculations of Models 1-6, respectively. As for the AIMD simulations, the time step was 2.0 fs. The structures used for AIMD calculation of χ-AgMg and C interface phases contained 512 and 464 atoms, respectively. The **k**-points grid for the AIMD simulations was set as 1 × 1 × 1 due to the large dimensions of the supercell. The melting temperature calculated by AIMD usually differs from the actual melting temperature[68,69]. Therefore, the melting temperature of pure Al was predicted through AIMD. The AIMD simulations were performed at different temperatures using 4 × 4 × 4 supercell (the total number of Al atoms is 256). The obtained radial distribution function and the structural evolution of pure Al at different temperatures were shown in Supplementary Figs. 13, 14, respectively. The results show that the predicted melting temperature of pure Al is between 1073 K (800 °C) and 1173 K (900 °C). Thus, the temperature used to simulate the structural evolution of the χ-AgMg and C interface phase by AIMD was set as 1073 K, which is close to the actual melting temperature of Al. In addition, the

formation energy per atom was calculated using the following Eq. (4):

$$\Delta E_{ss}^{form}(ModelX) = \frac{E_t - \Sigma N_i E_i}{N} \qquad (4)$$

where $E_t$ is the total energy of supercell model, $N_i$ ($i$ = Al, Cu, Mg, Ag, and Si) is the number of each type of atom in the model, $E_i$ represents the energy per atom of pure Al, Cu, Mg, Ag, or Si in their standard states, respectively. $N$ is the total number of atoms in the model.

## Data availability

The datasets generated and/or analyzed during the current study are available from the corresponding author on request. Besides, the raw TEM images, different line profile data and raw DFT data used in this study are available on the Mendeley Data website at https://data.mendeley.com/datasets/gmmzj9k38w/draft?a=406d89dc-360a-4fc1-881e-61330047989a.

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

## Acknowledgements

This work is financially supported by the National Natural Science Foundation of China (52071340 and 51820105001) and Natural Science Basis Research Plan in Shaanxi Province of China (2021JM-203). The authors thank the Hunan Center for Electron Microscopy, Electron Microscopy Center of Powder Metallurgy Research Institute and the High-Performance Computing Center at CSU for assistance in TEM experiments and computations. The first author sincerely appreciates his fiancée Miss. Min Yuan for her support and company.

## Author contributions

K.L. and Y.D. initiated and supervised the project. Q.L. conceptualized, designed, and prepared the alloys, performed most of the experiments and the DFT calculations, and wrote the manuscript. J.W. performed the DFT calculations. H.L., S.J., and G.S. performed the APT experiments and data analysis. J.L. and L.W. provided assistance in HAADF examinations. B.J. and X, L contributed to the construction of the crystal structure of the newly discovered χ-AgMg interface phase. All authors extensively discussed the data. K.L., G.S., J.W., Y.D. and L.L. revised the manuscript.

## Competing interests

The authors declare no competing interests.
