## [Peer Review File · Nature Communications]

Synergy of multiple precipitate/matrix interface structures for a heat resistant high-strength Al alloyREVIEWER COMMENTS

Reviewer #1 (Remarks to the Author):

This manuscript reports the design and characterisation of a novel high-strength aluminium alloy with improved thermal stability compared to existing alloys. This topic of designing aluminium alloys that can retain their high strength at higher temperatures is perhaps one of the currently most important ones in the field. Based on what the authors describe as strength retention ratio, their new alloy exhibits a significant improvement over similarly Al-Cu based alloys. This, in itself, is an important result. This improvement is ascribed to special interfacial structures, with phases segregating at both coherent and semicoherent interfaces. According to the authors, this slows down precipitate lengthening and thickening and therefore precipitate coarsening, the main reason for the decrease in yield strength at elevated temperatures.

This paper is of high quality experimentally and computationally, so I do not dispute its results. However I have concerns about the authors' interpretation of the results.

One of my main concerns is with the claim that "enveloping precipitates with multiple interface phases" is a new concept. It is indeed common for precipitates to be attached to other precipitates and have alloying elements segregating at their interfaces, whether in 6xxx series alloys or in Al-Cu-based alloys. In addition, the semicoherent interfaces do not seem to be fully covered, but rather connected to other precipitates. The atom probe data shown in Fig. 2(g), for instance, clearly shows that the precipitate is not enveloped but partially covered.

Furthermore, the authors do not present sufficiently convincing arguments that the improved thermal stability of the new alloy compared with other Al-Cu alloys is due to the interfacial structures and not other factors (eg different solute concentrations, different precipitate number densities at peak hardness...etc). A more rigorous analysis needs to be presented for the authors to claim that the presence of interfacial phases is the cause of the hindered precipitate coarsening.

Also problematic is the unclear link with alloy design. The authors seem to imply that they designed this alloy so as to display special interfacial covering (unless I misunderstood). But it is not clear how this could be so, since one of the interfacial phases was characterised once the alloy was made, so it could not be known that this phase existed at the alloy design stage. The authors need to be clearer about the alloy design procedure, which is one of the strengths of this paper.

I understand the challenges of designing an alloy based on bulk phase data, as pointed out by the authors. Their description of how they went about selecting the alloy composition is actually quite nice, and it is a pity this is reserved for the supplementary material. Given the authors' claims about designing the alloy, their CALPHAD procedure should be summarised in the main text of the paper.

There are also several alloy design criteria I do not understand:

- Why suppress the omega phase? It is well known to be an efficient strengthening precipitate.
- Why slow down precipitate lengthening? It is not clear how this would lead to a decrease in strength.
- Can the authors indicate how much the alloy composition can change without major effects to their

observations?

-

Other points:

- Page 4: heading should be “Results”.
- Page 4: definition of strength retention ratio?
- Error bars must be included in Fig. 1(f)-(g).
- Caption of Fig. 1: “more details can be found in Supplementary Table” – where?
- Error bars must include only one significant figure (eg 43 ± 20 nm on page 5, line 3).
- Some crystallographic information would help in Fig. 2(g) – it is hard to visualise.
- How were the concentrations determined using EDX? (Fig. 3)
- Page 7, line 24, Fig. 3c should be Fig. 3d.
- Page 8: is the epsilon phase a new phase or is it known?
- Page 8: FFT is a Fast Fourier Transform, not transformation (or an FFT pattern, as written later).
- Pages 8-9: why is only HRTEM of the epsilon phase shown? HAADF-STEM images should be easier to interpret.
- Page 13: “In the early stage of artificial ageing process, the disordered L phase can serve as heterogeneous nucleation sites to promote the nucleation of the θ' , thereby reducing its size and increasing the number density. On the other hand, due to the high thermal stability of disordered L phase, the broadening of θ' at semi-coherent interface can be hindered”. Any evidence to support these assertions?
- Page 16: Was any Fourier filtering performed on the STEM images?

Reviewer #2 (Remarks to the Author):

The manuscript describes thorough structural and property characterization of an Al-Cu-Mg-Ag-Si-Sc alloy which shows high heat resistance and high strength. The English is quite good, and the paper presents a huge amount of work, both experimental (TEM (STEM, EDX, HRTEM) and APT) and from modelling and simulations (CHALPAD, DFT, MD and QSTEM). I think the paper is worthy to be published in Nature communications, but it needs a few improvements and clarifications, as commented on here:

When it comes to heat resistant Al alloys in general, when introduced in the introduction, I think you could refer to the paper by Marioara et al (<https://doi.org/10.1007/s11661-014-2250-0>) where it is shown how manipulation with the microstructure can change thermal stability of Al alloys.

It is unclear to me (not explained well?) what is the difference between the ‘newly discovered’ AgMg phase and the ‘thick’ AgMg (Ag_5Mg_3) phase called ξ phase. (This is partly commented in the paper, but no conclusions.) Is the former one a thin interface phase? Is the structure of the ‘newly discovered’ one the same as the thick one/or parts of it? And is this ‘newly discovered’ phase discovered in this paper? You say it is not the same that has been reported earlier by the Rosalie and Kang papers, but it is not easy to understand the difference? Why do you not upload the structure file?

Why has thermal exposure been done at both 200°C and 210°C? Would it be possible to explain why two temperatures are used and what would be the difference?

It is a lot of details in the TEM figures 3 and 4. Would it be possible to remove a few of the details and make the figures larger and a bit more 'digestive'? For example, the EDX results in 3a) should be presented clearer. They represent an important result and should be better and clearer presented. The QSTEM simulations should be a bit larger. Maybe move something into the supplementary results? The HRTEM results and the fft in 4 g) and h) could be discussed more thorough in the result part.

I feel that the discussion and comparison between TEM and APT should be extended. Traditionally, APT is best to determine chemical composition (and TEM the structure), but here it seems like the EDX results are what is used to find the chemical composition of the phase? As it is now, I don't see what results and use have come out of the APT. Please explain what use you have had from APT in this work.

Why are the modelling results presented in the discussion part? Wouldn't it be better to present them in the results part?

When 'AIMD' is used in the text for the first time the abbreviation should be written out.

The AIMD results are obtained with a temperature of 1073K. This is well above the melting temperature of Aluminium. It is said that this is to increase the speed, but the reliability of this is not discussed. I would think that the results obtained in these calculations will not be reproducible, as the structure is gone? Could you please give a comment on this?

It is said that ThermoCalc is used in this work – Are all the phases which exist in this alloy formed in this model (with 6 elements) included in the thermodynamic calculations? I see that some design principles/advice are given; will these calculations be possible to reproduce?

How accurate are the EDS compositions and atomic fractions found from figure 3d and e? It would be nice if some error /std could be given for these numbers... (see for example this paper: <https://doi.org/10.1016/j.micron.2017.02.007>)

Response to reviewers' comments

Dear Editor and Reviewers,

The authors would like to thank you for the time and efforts that you spent on reviewing our manuscript. We gratefully appreciate for your valuable comments. All these comments are carefully considered and responded accordingly in the following section. Revisions are marked in **yellow background** in the manuscript and also in this response letter.

We sincerely hope these response and revisions can satisfy you.

Regards,

Yours sincerely,

Kai Li and Yong Du, on behalf of all authors

REVIEWER COMMENTS

Reviewer #1 (Remarks to the Author):

This manuscript reports the design and characterisation of a novel high-strength aluminium alloy with improved thermal stability compared to existing alloys. This topic of designing aluminium alloys that can retain their high strength at higher temperatures is perhaps one of the currently most important ones in the field. Based on what the authors describe as strength retention ratio, their new alloy exhibits a significant improvement over similarly Al-Cu based alloys. This, in itself, is an important result. This improvement is ascribed to special interfacial structures, with phases segregating at both coherent and semicoherent interfaces. According to the authors, this slows down precipitate lengthening and thickening and therefore precipitate coarsening, the main reason for the decrease in yield strength at elevated temperatures.

This paper is of high quality experimentally and computationally, so I do not dispute its results. However I have concerns about the authors' interpretation of the results.

One of my main concerns is with the claim that “enveloping precipitates with multiple interface phases” is a new concept. It is indeed common for precipitates to be attached to other precipitates and have alloying elements segregating at their

interfaces, whether in 6xxx series alloys or in Al-Cu-based alloys. In addition, the semicoherent interfaces do not seem to be fully covered, but rather connected to other precipitates. The atom probe data shown in Fig. 2(g), for instance, clearly shows that the precipitate is not enveloped but partially covered.

Response: Thanks for your suggestions. We agree with you that the word “new” is a bit controversial in our case. Thus, in the abstract we have revised the sentence “This concept of enveloping precipitates with multiple interface phases and segregation layers provides a new strategy for designing other heat resistant materials” into “...provides an effective strategy for designing other heat resistant materials”. We have also deleted the word “novel” in the title “Novel synergy of multiple precipitate/matrix interface structures...” as this word is not allowed according to the policy of *Nature Communications*, although the structure was indeed not seen before. The shortened title is now "Synergy of multiple precipitate/matrix interface structures for a heat resistant high-strength Al alloy"

In this paper, the multiple interface structures include partially distributed **interface phases** such as C, L and the newly discovered AgMg phase, and Sc **segregation layers** surrounding the precipitates, which jointly envelope the θ' precipitates. In addition, the atom probe tomography data shown in Fig. 2h indicates that the interfacial positions without interface phases are covered by the Sc segregation layers. This is also reflected in Supplementary Fig. 1 where more Sc can be found at the interfaces. Thus, the precipitates are nearly completely enveloped.

Furthermore, the authors do not present sufficiently convincing arguments that the improved thermal stability of the new alloy compared with other Al-Cu alloys is due to the interfacial structures and not other factors (eg different solute concentrations, different precipitate number densities at peak hardness...etc). A more rigorous analysis needs to be presented for the authors to claim that the presence of interfacial phases is the cause of the hindered precipitate coarsening.

Response: In order to visualize the inhibitory effect of multiple interface structures on the coarsening of the precipitates, a control alloy without these structures was designed. The results of previous studies showed that Ag¹, Sc² and Si^{3,4} elements could participate in the formation of interface structures at the precipitate/matrix interface. Besides, the addition of Mg will promote the formation of S-Al₂CuMg precipitates.

Therefore, the alloy composition of the control alloy is set as Al-4Cu (in wt.%).

Supplementary Figure 2. **The mechanical properties of the Al-4Cu alloy.** **a** Microhardness of the Al-4Cu alloy aged for different times at 175°C. **b** Stress-strain curves of the peak aged Al-4Cu alloy before and after thermal exposure.

Supplementary Figure 3. **Changes in the size of the θ' -Al₂Cu precipitates in the peak aged Al-4Cu alloy before and after thermal exposure heat treatment.** **a** and **b** Distributions of thickness and diameter of the Al-4Cu alloy, for both the peak-aged and thermally exposed states. **c** and **d** Typical images used to measure the diameter and thickness of θ' -Al₂Cu precipitates in the peak-aged state of the current alloy. **e** and **f** Typical images for the state after thermal exposure.

Supplementary Fig. 2 shows the mechanical properties of the control alloy under different conditions. The microhardness results indicate that the Al-4Cu alloy reaches peak aging after aging at 175°C for 16 h. In addition, the yield strength values of the peak aged Al-4Cu alloy before and after thermal exposure are 208 ± 12 and 137 ± 3 MPa, respectively. The strength retention ratio of the Al-4Cu alloy is 65.9%, which

indicates that the strength of the alloy decreases significantly after thermal exposure at 210°C for 100 h. Furthermore, rapid coarsening of the precipitates was observed as shown in Supplementary Fig. 3. According to Supplementary Figs. 3a-b, the average thickness values of θ' -Al₂Cu before and after thermal exposure are 1.1 ± 0.4 nm and 10 ± 3.5 nm, while those values for diameter are 25 ± 10 nm and 318 ± 210 nm, respectively. The statistical results show that θ' -Al₂Cu precipitates without interface structures have been obviously coarsened after thermal exposure, thereby leading to the rapid decrease in tensile properties of the Al-4Cu alloy.

We have added two sentences in lines 9-14 page 7: “In comparison, an Al-4Cu control alloy has also been prepared in the same way as for the current Al-4Cu-0.315Mg-0.5Ag-0.21Si-0.09Sc alloy. As can be seen in Supplementary Figs. 2-3, the unsatisfactory thermal stability of the peak aged Al-4Cu alloy intuitively reveals the crucial role the multiple interface structures plays in effectively hindering the coarsening of θ' -Al₂Cu precipitates in the currently alloy.”

As for the effect of solute concentrations in the precipitates, it should be noted the main precipitates in both the Al-4Cu-0.315Mg-0.5Ag-0.21Si-0.09Sc alloy and the Al-4Cu control alloy are θ' -Al₂Cu. As these two alloys have reached the peak aging state before thermal exposure heat treatment, the Cu solutes in both alloys should have been mostly used to form the precipitates. The contents of remaining Cu solutes in matrix of the two alloys are similar under the peak aging condition. Therefore, the Cu solute concentrations in Al matrix of both the Al-4Cu alloy and the currently designed alloy should have no significantly different effects on the coarsening behavior of precipitates. However, we have to admit that with the additions of Mg, Si, Ag and Sc, the type, size and number density of precipitates change, and this will indeed influence the strength and heat resistance of the alloy.

The high number density as well as uniform size distribution of precipitates in the current Al-4Cu-0.315Mg-0.5Ag-0.21Si-0.09Sc alloy, as shown in Fig. 5a and Supplementary Fig. 1, probably acts as one of the factors in hindering the coarsening of the precipitates. This has been validated in the work of Makineni et al.⁵ where Nb and Zr additions increased the number density of θ' and the heat resistance of their Al-2Cu-0.1Nb-0.15Zr alloy. However, in our work, by comparing Supplementary Fig. 3 with Fig. 1, it can be found the θ' precipitates are even finer and the number density is probably significantly higher in the Al-4Cu control alloy processed in the same way as the current alloy, while its heat resistance is much worse. Thus, it is believed the

presence of the multiple interface structures at the θ' /Al interface is the main cause of effectively inhibiting the coarsening of θ' precipitates in the current alloy. This paragraph has been added to the paper in lines 1-11 of page 15.

Also problematic is the unclear link with alloy design. The authors seem to imply that they designed this alloy so as to display special interfacial covering (unless I misunderstood). But it is not clear how this could be so, since one of the interfacial phases was characterised once the alloy was made, so it could not be known that this phase existed at the alloy design stage. The authors need to be clearer about the alloy design procedure, which is one of the strengths of this paper.

Response: Thanks for the suggestion. In the alloy design, we indeed intended to get multiple interface structures at the θ' /Al interface including Sc segregation layers, C-AlMg₄Si₃Cu interface phase and the AgMg segregation layers. As introduced in the part of introduction in the manuscript (as shown in page 3, lines 15-18), Gariboldi et al.⁶ reported that Mg and Ag solutes in Al-Cu-Mg-Si-Zr-Ti-Ag alloy could segregate at the θ' /Al interface, thus forming an AgMg segregation layers at the θ' /Al interface. However, there is currently no research report on the composition of the AgMg layer at the θ' /Al interface. Therefore, during the design process, the AgMg segregation layer at the Ω /Al interface as reported in literature was used as a reference to control the content of Mg element, which was used to form the AgMg segregation layers at the θ' /Al interface. Indeed, we just expected to obtain the AgMg segregation layers but did not know what their atomistic structures will be. Thus, we have added a sentence (as shown in page18, lines 8-10) “The AgMg segregation layers were expected, although during the design it was not known what their atomistic structures would be” in the “*Alloy design by CALPHAD*” section. The same thing holds true for the C interface phase. This section, containing less than 1000 words, is now directly put into the *Methods* section of the paper, since placing the alloy composition design process in the supplementary file is believed to weaken the strength of the paper.

I understand the challenges of designing an alloy based on bulk phase data, as pointed out by the authors. Their description of how they went about selecting the alloy composition is actually quite nice, and it is a pity this is reserved for the supplementary material. Given the authors’ claims about designing the alloy, their

CALPHAD procedure should be summarised in the main text of the paper.

Response: Thanks for your encouragement and suggestion. As mentioned in the response to the previous question, the alloy design process has been placed in the “*Methods - Alloy design by CALPHAD*” section of the manuscript. Meanwhile we have added the following summarizing sentences in the beginning of *Results* section.

"The calculated phase equilibria at the homogenization/solution temperature of 540°C and the thermal exposure temperature of 210°C guided the optimization of solute concentrations step by step, under the principles of: (a) avoiding detrimental AlCuSc phases while introducing Sc for the formation of Sc-rich segregation layers on precipitates, (b) inhibiting the formation of detrimental S-Al₂CuMg phase as well as the Ω precipitate which is not heat resistant enough, and (c) promoting the possible formation of C/L phase and AgMg-rich interface structures at precipitate/matrix interfaces. The CALPHAD details are introduced in the *Methods* section."

We hope the editor and the reviewer could understand our choice, since without the CALPHAD details given in the *Methods* section these sentences would be difficult to understand for readers.

There are also several alloy design criteria I do not understand:

- **Why suppress the omega phase? It is well known to be an efficient strengthening precipitate.**

Response: Ringer et al.⁷ and Hutchinson et al.⁸ reported that the size stability of the Ω phase decreased significantly when the thermal exposure temperature was higher than 200°C. Hutchinson et al.⁸ found that when the thermal exposure temperature was higher than 200°C, there were many ledges with 1/2 Ω unit cell in height at the Ω /Al interface. The ledges at the interface will increase the thickening kinetics of Ω phase, leading to the coarsening of the Ω phase above 200°C. Therefore, we expect to obtain the multiple interface structures with better stability than the Ω phase by alloy design using CALPHAD. This is also the reason why the thermal exposure temperature of 210°C was chosen at the design process.

In the meantime, the reason why Ω is suppressed are also added in the manuscript as shown below (page 2, line 29; page 3, lines 1-7):

"Hutchinson et al.⁸ found that the AgMg segregation layers could hinder the

coarsening of the precipitates due to the retarded diffusion of solutes when the thermal exposure temperature was below 200°C, thereby improving the thermal stability of the alloy. However, when the thermal exposure temperature was higher than 200°C, the ledges with 1/2Ω unit cell in height were more likely to form at the Ω/Al interface, which would increase the thickening kinetics of Ω phase. That is, such a single segregation structure at the precipitates/matrix interface still cannot effectively hinder the coarsening of the precipitates."

• **Why slow down precipitate lengthening? It is not clear how this would lead to a decrease in strength.**

Response: Nie et al.⁹ investigated the effect of precipitate number density and aspect ratio on critical resolved shear stress (CRSS) increments for θ'-Al₂Cu precipitate plates. The results (shown as Fig.10 in their paper and also shown below) showed that increasing the aspect ratio of θ'-Al₂Cu precipitates could improve the mechanical properties of the alloy when the number density of θ'-Al₂Cu precipitates was the same,

Fig.10 in the paper of Nie et al. **Effects of precipitate number density and aspect ratio on CRSS increments for θ' precipitate plates at a volume fraction of 0.025.**⁹

while the mechanical properties of the alloy could also be improved by increasing the

number density of θ' -Al₂Cu precipitates when the aspect ratio of θ' -Al₂Cu precipitates was the same. However, a decrease in the number density of precipitated phases has a more obvious effect on the alloy mechanical properties. That is, even though the increase of the θ' -Al₂Cu precipitates length can improve the alloy mechanical properties to some extent, the lengthening of θ' -Al₂Cu precipitates will also lead to the obvious decrease of the number density of θ' -Al₂Cu precipitates, leading to a decrease in the alloy properties. Therefore, limiting the lengthening of the θ' -Al₂Cu precipitates during thermal exposure process can also improve the strength retention ratio of the alloy.

Thus, we have added the sentence "broadening of θ' -Al₂Cu precipitates will also lead to the obvious decrease of the number density of θ' -Al₂Cu precipitates, leading to a decrease in the alloy properties according to the work of Nie et al." in lines 16-18, page 13.

• **Can the authors indicate how much the alloy composition can change without major effects to their observations?**

Response: According to our previous studies, the content of Mg has a significant effect on the heat resistance of the alloy. Increasing the content of Mg will change the type of precipitates during the heat exposure and lead to a decrease in the heat resistance of the alloy. It is important to ensure both the promotion of C/L phase generation and the availability of additional Mg elements for the generation of AgMg segregation layers when designing the alloy composition. As shown in Fig. R1, when the composition falls into the (Al)+ θ +Q region, there is no extra Mg in the alloy for the generation of AgMg segregation layers. When the composition falls into the Al+ θ +Mg₂Si phase area, it indicates that the content of Mg in the alloy is too high, which will easily lead to the subsequent phase transformation of the C/L precipitates towards Mg₂Si, thus leading to the decline in the thermal stability of the alloy. Therefore, the content of Mg should be in the range of 0.25 wt.% to 0.35 wt.% based on the Fig. R1. Then, according to the relationship between the Mg/Si/Ag alloy element contents described in the "*Alloy design by CALPHAD*" section of the manuscript, the content of Si should be in the range of 0.16 wt.% to 0.23 wt.%, while the content of Ag should be between 0.45 wt.% and 0.58 wt.%. The contents of Cu and Sc can be set according to the ternary isothermal section of Al-Cu-Sc system at 540°C shown in Supplementary Fig. 10b. When the content of Cu is too low, the number density and volume fraction of θ' -Al₂Cu precipitates will be decreased, which reduces the mechanical properties of

the alloy. When the content of Cu is too high, the formation of AlCuSc phase will reduce the Sc content in the supersaturated solid solution, leading to a reduction in the Sc segregation layers and reducing the thermal stability of the alloy. Therefore, the content of Cu should be in the range of 3.5 wt.% to 4.5 wt.%, while the content of Sc should be between 0.08 wt.% and 0.11 wt.%.

Thus, the composition range is summarized as: Al-(3.5~4.5) Cu-(0.25~0.35) Mg-(0.45~0.58) Ag-(0.16~0.23) Si-(0.08~0.11) Sc (in wt.%). We prefer not adding it into the main text of the paper as we do not want to mislead the readers. This is just our estimation.

Figure.R1 Pseudo-ternary isothermal section of Al-Cu-Mg-0.2Si alloy at 210°C.

Other points:

- **Page 4: heading should be “Results”.**

Response: Thanks. The mistake has been corrected in the revised manuscript.

- **Page 4: definition of strength retention ratio?**

Response: The strength retention ratio is defined as the ratio (in percentage) between the strength of a peak-aged alloy after thermal exposure and the peak-aged strength. It has now been defined as “strength retention ratio (the ratio between the yield strength values of an alloy after and before a thermal exposure)” at its first appearance in page 4, lines 19-20.

- **Error bars must be included in Fig. 1(f)-(g).**

Response: Thanks for your suggestion. Figs. 1f-g show the histogram of the frequency distribution of the diameter and thickness of the θ' -Al₂Cu precipitates. The

relative frequency in Figs. 1f-g is a specific statistical data obtained from the measured data, and it is not possible to calculate the error bars. However, in the calculation of the average diameter and thickness of the θ' -Al₂Cu precipitates, the corresponding error has been calculated based on the measured data. Also, the label of y-axis in Figs. 1f-g is modified to the relative frequency.

• **Caption of Fig. 1: “more details can be found in Supplementary Table” – where?**

Response: Sorry for the inconvenience of reviewing this manuscript. As this table is more than one page, it needs to be submitted as an individual Excel file and cannot be added directly to the *Supplementary Information* file. Therefore, in the last version, the table of mechanical properties of different alloys was submit as an individual supplementary material in the form of Excel. At present, the table has been revised and added to the *Supplementary Information* file.

• **Error bars must include only one significant figure (eg 43±20 nm on page 5, line 3).**

Response: Thanks for your suggestion, the mistake has been corrected in the revised manuscript, e.g. from “42.8 ± 17.8 nm and 45.6 ± 19.3 nm” to “43 ± 20 nm and 46 ± 20 nm” in page 4, lines 25-26.

• **Some crystallographic information would help in Fig. 2(g) – it is hard to visualise.**

Response: Thanks for your suggestion. Fig. 2g shows the results of the APT detection, with different colors representing the different elements in the precipitates. For example, purple represents Mg element and green represents Ag element, etc. Therefore, the green phase containing Ag and Mg elements in Fig. 2g represents the AgMg interface phase at the θ' /Al interface, and its structure is consistent with that shown in Fig. 2c. The purple phase containing Al, Mg, Si and Cu elements at the coherent θ' /Al interface represents the C interface phase, and its structure is consistent with that shown in Fig. 2d. The purple phase at the semi-coherent θ' /Al interface represents the disordered L interface phase, and its local structure is similar with that shown in Fig. 2d for the ordered C phase. Since there are many annotations in Fig. 2g, adding some crystallographic information will make the picture more crowded, hence crystallographic information was not added in Fig. 2g.

In order to make Fig. 2g more friendly to readers, we have revised a sentence in page 7 into “Fig. 2g also shows the relationship between the interface phases and the θ' -Al₂Cu, demonstrating that the AgMg (in green color) and lath-like C (in purple color) interface phases mainly exist at the coherent θ' /Al interface, while there is only

disordered L phase (also in purple color) at the semi-coherent θ'/Al interface”.

Meanwhile, to facilitate the understanding of the structures of C interface phase and disordered L phase in Figs. 2d-f, the crystal structure of C phase in Figs. 2d has been enlarged a bit. Furthermore, we have now uploaded all crystallographic information files (cif) with the paper.

• **How were the concentrations determined using EDX? (Fig. 3)**

Response: The concentration line profile of Mg is a line profile of the entire AgMg interface phase using the IntensityProfile tool in the Velox software in the raw EDX elemental maps file. This has been added to the *Methods* section of the manuscript. As mentioned in the figure caption, the line profile is with an integration width of 7 nm, which means each composition in the profile is an average composition.

• **Page 7, line 24, Fig. 3c should be Fig. 3d.**

Response: Thanks very much. The mistake has been corrected in the revised manuscript.

• **Page 8: is the epsilon phase a new phase or is it known?**

Response: According to the updated HAADF-STEM image, the corresponding FFT pattern and EDX maps in Fig. 4, the composition of ξ phase was $\text{Ag}_1\text{Mg}_{1-x}\text{Al}_x$ ($x = 0.5$) and the crystal structure is B2 (based on BCC). There are some structures of Ag_1Mg_1 among the available structures with BCC crystal structure, but these available structures do not contain Al element. The EDX result shows that Mg and Al atoms take one site in the unit cell together in the ξ phase. Therefore, the ξ phase found in our work is a new phase, and the lattice parameters of ξ - $\text{Ag}_1\text{Mg}_{1-x}\text{Al}_x$ ($x = 0.5$) phase was determined as: space group $Pm-3m$, $a = b = c = 3.34 \pm 0.10 \text{ \AA}$ and $\alpha = \beta = \gamma = 90^\circ$. In addition, the crystal structure file is also uploaded as *Ksi phase-AgMgAl.cif*. We have accordingly updated the related text in page 9, lines 9-27 (not copied here as the text is too long).

• **Page 8: FFT is a Fast Fourier Transform, not transformation (or an FFT pattern, as written later).**

Response: Thanks for correcting our mistake. The mistake has been corrected in the revised manuscript.

• **Pages 8-9: why is only HRTEM of the epsilon phase shown? HAADF-STEM images should be easier to interpret.**

Response: Thanks for your suggestion. The HRTEM images has now been replaced by atomic resolution HAADF-STEM and STEM-EDX images (as shown in

Figs. 4d-f). According to these results, the lattice parameters of ξ -Ag₁Mg_{1-x}Al_x (x = 0.5) phase was determined as: space group *Pm-3m*, $a = b = c = 3.34 \pm 0.10$ Å and $\alpha = \beta = \gamma = 90^\circ$.

• **Page 13: “In the early stage of artificial ageing process, the disordered L phase can serve as heterogeneous nucleation sites to promote the nucleation of the θ' , thereby reducing its size and increasing the number density. On the other hand, due to the high thermal stability of disordered L phase, the broadening of θ' at semi-coherent interface can be hindered”. Any evidence to support these assertions?**

Response: Liu et al.¹⁰ reported that Si addition to Al-Cu-Mg alloy could promote the formation of Q''-type precipitates (also could be named as L phase). Their TEM images confirmed that Q''-type or L precipitates could serve as heterogeneous nucleation sites to promote the nucleation of the θ' . In addition, Marioara et al.¹¹ reported that the fine lath-shaped, Cu-containing, disordered L phase had a high thermal stability, which could promote the heat resistance of the Al alloys. Therefore, during thermal exposure process, the L phase segregating at the semi-coherent interface of θ' can stabilize the interface and hinder the broadening of θ' .

Therefore, for the two sentences mentioned by the reviewer, we have added one literature for each sentence.

• **Page 16: Was any Fourier filtering performed on the STEM images?**

Response: Each atomic resolution HAADF image of peak aged alloy was acquired by superimposing 50 fast scan images recorded with the drift corrected frame integration (DCFI) technology. Thus, the atomic resolution HAADF images of peak aged alloy shown in Figs. 2-3 were not filtered. The atomic resolution HAADF images of alloy after thermal exposure were filtered using Gaussian blur or Radial Wiener, while all the atomic resolution EDX images were filtered using Radial Wiener in Velox software. The atomic arrangement and the intensity variations of the atomic columns in the filtered image are consistent with the original image. This has been added to the *Methods* section of the manuscript. As described in the *Data availability* section, we have uploaded all raw images (unfiltered) in the following link:
<https://data.mendeley.com/datasets/gmmzj9k38w/draft?a=406d89dc-360a-4fc1-881e-61330047989a>

Reviewer #2 (Remarks to the Author):

The manuscript describes thorough structural and property characterization of an Al-Cu-Mg-Ag-Si-Sc alloy which shows high heat resistance and high strength. The English is quite good, and the paper presents a huge amount of work, both experimental (TEM (STEM, EDX, HRTEM) and APT) and from modelling and simulations (CHALPAD, DFT, MD and QSTEM). I think the paper is worthy to be published in Nature communications, but it needs a few improvements and clarifications, as commented on here:

When it comes to heat resistant Al alloys in general, when introduced in the introduction, I think you could refer to the paper by Marioara et al (<https://doi.org/10.1007/s11661-014-2250-0>) where it is shown how manipulation with the microstructure can change thermal stability of Al alloys.

Response: Thanks for the recommendation. We have referred to the relevant literature and made some modifications in the *Introduction* section, proposing to improve the heat resistance of the alloy by increasing the content of heat-resistant precipitates. The revised parts in page 3, lines 11-15 are shown below:

Marioara et al.¹¹ reported that the thermal stability of 6xxx alloys could be improved by reasonably controlling the contents of Mg, Si and Cu to form the fine lath-shaped, Cu-containing, disordered L phase. Therefore, the disordered L phase can be added to Al-Cu-based alloys as a heat-resistant precipitate to improve the heat resistance of Al-Cu-based alloys.

It is unclear to me (not explained well?) what is the difference between the ‘newly discovered’ AgMg phase and the ‘thick’ AgMg (Ag₅Mg₃) phase called ξ phase. (This is partly commented in the paper, but no conclusions.) Is the former one a thin interface phase? Is the structure of the ‘newly discovered’ one the same as the thick one/or parts of it? (Q1) And is this ‘newly discovered’ phase discovered in this paper? (Q2) You say it is not the same that has been reported earlier by the Rosalie and Kang papers, but it is not easy to understand the difference? Why do

you not upload the structure file? (Q3)

Response to *Q1*: Sorry for our misleading description that causes inconvenience in your reviewing of this manuscript. To illustrate more obviously the atomic arrangement of the thick AgMg phase as well as the structural information, the HRTEM image and the corresponding FFT pattern are now replaced by atomic resolution HAADF-STEM image and the corresponding FFT pattern (as shown in Figs. 4d-e). According to the updated HAADF-STEM image, the corresponding FFT pattern and EDX maps, the composition of ξ phase was $\text{Ag}_1\text{Mg}_{1-x}\text{Al}_x$ ($x = 0.5$) and the crystal structure is BCC and the lattice parameters of ξ phase was determined as: space group $Pm-3m$, $a = b = c = 3.34 \pm 0.10 \text{ \AA}$ and $\alpha = \beta = \gamma = 90^\circ$. Therefore, the thick AgMg phase named ξ phase is different from the newly discovered AgMg interface phase. In addition, the crystallographic information file is also uploaded as cif files named *Ksi phase-AgMgAl.cif*.

Response to *Q2*: As shown in the *Introduction* section (page 3, line 15-18), The work of Gariboldi et al.⁶ reported that Mg and Ag solutes in Al-Cu-Mg-Si-Zr-Ti-Ag alloy could segregate at the θ'/Al interface. However, the specific structure as well as the composition of the AgMg segregation layers was not characterized. Whether this segregation layer has an ordered structure or not is unknown. Therefore, the AgMg interface phase with three atomic layers was characterized for the first time, and the model was constructed for the first time in this paper.

Response to *Q3*: Fig. R2 shows the HAADF-STEM images and structure of AgMg bi-layer reported by Kang et al.¹². and Ag segregation layer reported by Rosalie et al.¹. According to Fig. R2, the structures of the AgMg bi-layer and Ag segregation layer are both different from that of the newly discovered AgMg interface phase in the current alloy. Meanwhile, to visually compare the newly discovered AgMg interface phase with the AgMg bi-layer at the Ω/Al interface reported by Kang et al.¹² and the Ag segregation layer at the θ'/Al interface reported by Rosalie et al.¹, the structure of these three AgMg or Ag segregation structures were shown in Supplementary Fig. 6, while the crystallographic information of these three structures were also uploaded as supplementary files (*Al2Cu-AgMg-Al.cif*, *Omega-AgMg_Kang.cif*, *Al2Cu-Ag-Al_Rosalie.cif*).

Figure. R2 The HAADF-STEM images and structure of AgMg or Ag segregation layer at the interface of Ω and θ' . a-d AgMg bi-layer at the Ω /Al interface along different direction¹². e-f Ag segregation layers at the θ' /Al interface along different direction¹. g Structure containing AgMg bi-layer¹³. h Structure containing the double atomic layer Ag segregation layer¹.

Supplementary Figure 6. **Different AgMg segregation structures.** a Structure of the uniformly distributed AgMg segregation layer containing double atomic layers at the Ω /Al interface reported by Kang et al.^{12, 13}. b Structure of the Ag segregation layer with double atomic layers at the θ' /Al interface reported by Rosalie et al.¹. c Structure of the newly discovered AgMg interface phase at the θ' /Al interface in this work.

We have thus updated the paragraphs concerning the structures of the newly discovered AgMg interface phase and the newly discovered ξ -Ag₁Mg_{1-x}Al_x ($x = 0.5$) phase in page 9.

Why has thermal exposure been done at both 200°C and 210°C? Would it be

possible to explain why two temperatures are used and what would be the difference?

Response: At the beginning of the design, we expect to obtain the multiple interface structures with better stability than the Ω phase by alloy design using CALPHAD. Hutchinson et al.⁸ found that when the thermal exposure temperature was higher than 200°C, there were many ledges with $1/2\Omega$ unit cell in height at the Ω/Al interface. The ledges at the interface will increase the thickening kinetics of Ω phase, leading to the coarsening of the Ω phase above 200°C. Therefore, the thermal exposure temperature was initially chosen to be 210°C. However, when comparing the properties of different Al alloys after thermal exposure (as shown in Fig. 1a in manuscript), most of the Al-Cu based alloys in literature were thermally exposed at 200°C. Therefore, we also added a group of samples thermally exposed at 200°C.

We have added a sentence in the *Methods - Sample preparation* section: "The former temperature was selected to test the heat resistance in a higher temperature than that usually used, while the latter temperature was selected for comparison with available literature data."

It is a lot of details in the TEM figures 3 and 4. Would it be possible to remove a few of the details and make the figures larger and a bit more ‘digestive’? For example, the EDX results in 3a) should be presented clearer. They represent an important result and should be better and clearer presented. The QSTEM simulations should be a bit larger. Maybe move something into the supplementary results? The HRTEM results and the fft in 4 g) and h) could be discussed more thorough in the result part.

Response: Thanks for your suggestion. As shown below, the Figs. 3 and 4 had been revised. In addition, the HRTEM image and the corresponding FFT pattern were replaced by atomic resolution HAADF-STEM image and the corresponding FFT pattern (as shown in Figs. 4d-e).

In the meantime, Figs. 4d-f are discussed in detail as shown below (page 9, lines 9-27):

Figs. 4d-e show the atomic resolution HAADF-STEM images and the corresponding Fast Fourier Transform (FFT) pattern, respectively. As shown in the FFT pattern, the distance g_1 is 5.17 nm^{-1} , while those of g_2 and g_3 are 4.20 and 2.99 nm^{-1} ,

respectively. That is, the ratios g_1/g_3 and g_2/g_3 are 1.729 and 1.404, which are all consistent with the standard electron diffraction pattern of the body-centered cubic (BCC) structure along the [011] direction. In addition, the intensity line profile inserted in Fig. 4d shows that the light and dark atomic columns alternately arranged along A1 to A2, which could be identified as Ag and Mg+Al atomic columns according to the EDX mapping results shown in Fig. 4f. It is obvious that Mg and Al atoms jointly take one site in the unit cell, most probably in a disordered way, while Ag atoms solely occupy the other site. According to the information mentioned above, the B2 structure of ξ -Ag₁Mg_{1-x}Al_x ($x = 0.5$) phase has been constructed as inserted in Fig. 4d. The space group of the ξ phase is $Pm-3m$, while the lattice parameters are determined as $a = b = c = 3.34 \pm 0.10 \text{ \AA}$ and $\alpha = \beta = \gamma = 90^\circ$. The atomic ratio Ag: Mg: Al in ξ phase is 2: 1: 1, which is close to the ratio of about 51: 22: 28 as obtained from the EDX data (from the very thin area shown in Fig. 4d). Furthermore, the HAADF-STEM image (see the blue frame in Fig. 4d) simulated by QSTEM software¹⁴ along [011] direction of ξ phase is consistent with the experimental image. The crystallographic information file is uploaded as a supplementary file named *Ksi phase-AgMgAl.cif*.

Figure 3. The structure of the newly discovered AgMg interface phase. a-c and d Atomic resolution HAADF-STEM images of the AgMg phase along [100]_{Al} and [110]_{Al}, respectively. b Enlargement image of (a). e-h Atomic resolution EDX elemental maps of the area shown in (a). The intensity line profiles of the middle layer (L2) of the AgMg phase are inserted in (b) and (c). i Concentration line profile of Mg in the AgMg phase along the white arrow, the integration width is 7 nm. j 3D model of the θ /Al interface structure with the AgMg phase. It should be noted the viewing direction is parallel to [010] direction of the supercell in (j) for (a-b), [100] for (c) and [110] for (d). The HAADF-STEM images simulated by QSTEM using the constructed model along different directions are inserted in (a), (b) and (c).

Figure 4. HAADF-STEM images and EDX results of precipitates after thermal exposure at 210°C for 100 h. **a** HAADF-STEM image of θ' -Al₂Cu. **b** C interface phase at the θ' /Al interface. **c** Independently precipitated L phase. **d** Low-magnification and atomic resolution HAADF-STEM images of the ξ phase. **e** Corresponding FFT pattern of atomic resolution HAADF-STEM image in **(d)**. **f** EDX elemental maps of ξ phase. The unit cell of the ξ phase and the simulated HAADF-STEM image along $(011)_{\xi}$ by QSTEM have been inserted in **(d)**. The intensity line profiles inserted in **(d)** shows the intensity variation from A1 to A2.

I feel that the discussion and comparison between TEM and APT should be extended. Traditionally, APT is best to determine chemical composition (and TEM the structure), but here it seems like the EDX results are what is used to find the chemical composition of the phase? As it is now, I don't see what results and use have come out of the APT. Please explain what use you have had from APT in this work.

Response: Thanks for your suggestion. As APT cannot obtain atomic resolution chemical composition results, the atomic resolution EDX maps were used to identify the composition of the atomic columns of the AgMg interface phase. Meanwhile, the APT results had been used to illustrate the presence of the Sc segregation layers at the θ' /Al interface (as shown in Fig. 2h). Furthermore, to obtain a more accurate composition of the AgMg interface phase, all AgMg interface phases in the APT samples were counted, and the results are shown in Supplementary Fig. 5. The atomic ratio of AgMg interface phase detected by APT is about 1.44 (see lines 2-4 in page 8), which is close to that in the constructed crystal structure (shown in Fig. 3j).

Supplementary Figure 5. **The proxigram of total AgMg interface phase (at all interfaces) marked in pink.** The atomic ratio of AgMg interface phase is about 1.44.

Why are the modelling results presented in the discussion part? Wouldn't it be better to present them in the results part?

Response: Thanks for your suggestion. Your suggestion is absolutely reasonable. We have moved Fig. 5 and some related text from the *Discussion* section to the *Results* section. However, the purpose of conducting the modeling work is to discuss the effects of these various interface structures on thermal stability. We combine these results with discussion as there is a very tight connection between them, making the discussion more convenient. Therefore, we would prefer partially keeping this article structure and sincerely hope you can understand.

When 'AIMD' is used in the text for the first time the abbreviation should be written out.

Response: Thanks for your suggestion, the mistake has been corrected in the revised manuscript.

The AIMD results are obtained with a temperature of 1073K. This is well above the melting temperature of Aluminium. It is said that this is to increase the speed, but the reliability of this is not discussed. I would think that the results obtained in these calculations will not be reproducible, as the structure is gone?

Could you please give a comment on this?

Response: First of all, we are very sorry that there is a description error about the reason for choosing temperature. In fact, the melting temperature predicted by AIMD is usually differs from the actual melting temperature^{15, 16}. For example, we have predicted the melt temperature of pure Al through AIMD. The AIMD simulations were performed at different temperatures using 4×4×4 supercell (the total number of Al atom is 256). The obtained radial distribution function (RDF) and the structure evolution of pure Al at different temperatures are shown in Supplementary Figs. 12-13, respectively. The results show that the predicted melting temperature of pure Al is between 1073 K (800°C) and 1173 K (900°C). Thus, the simulation of structural evolution of the AgMg and C interface phase, we chose the temperature of 1073 K, which is close to the actual melting temperature of Al. This has been added to the *Methods* section of the manuscript (page 21, lines 14-23).

Supplementary Figure 12. **The radial distribution function (RDF) of pure aluminum at different temperatures.**

Supplementary Figure 13. The *ab initio* molecular dynamics (AIMD) results of pure Al calculated at different temperatures.

It is said that ThermoCalc is used in this work – Are all the phases which exist in this alloy formed in this model (with 6 elements) included in the thermodynamic calculations? I see that some design principles/advice are given; will these calculations be possible to reproduce?

Response: In fact, since the thermodynamic data of all metastable precipitates (C, L and the AgMg interface phase) except θ' are not available in the current database, the phases used for the thermodynamic calculations in this paper are all equilibrium phases. For example, the θ -Al₂Cu and Q-AlMgSiCu phases are the equilibrium phases θ' -Al₂Cu and C/L, respectively. The results of the thermodynamic calculations are reproducible according to the correlation between alloy composition, Gibbs energy description of phases, and phase equilibria (at a given temperature).

How accurate are the EDS compositions and atomic fractions found from figure 3d and e? It would be nice if some error /std could be given for these numbers... (see for example this paper: <https://doi.org/10.1016/j.micron.2017.02.007>)

Response: According to Supplementary Fig. 4, the components of six different AgMg interface phases detected by TEM were calculated using Velox. The average composition as well as the relative standard deviation was calculated. The average content of Ag in the AgMg phase is 61 ± 1 at.%, while that of Mg is 39 ± 1 at.%. The relative standard deviation is below $\pm 3\%$ of each averagely detected concentration, which indicates that the EDX results have a high stability. In addition, referring to the method proposed by Wenner et al.¹⁷ for calculating the signal-to-noise ratio (SNR), the residuals of the Gaussian fit are used as the noise and the fit height as the signal. The SNR of EDX maps shown in Fig. 3 is higher than 3, which indicates that the results of EDX maps are reliable. This has been added to the *Methods* section of the manuscript.

Supplementary Figure 4. **The EDS results of different AgMg interface phases.** The average content of Ag in the AgMg phase is 61 ± 1 at.%, while that of Mg is 39 ± 1 at.%. The average atomic ratio of Ag to Mg in the AgMg interface phase is about 1.56.

Reference mentioned in this response letter:

1. Rosalie, J. M.&Bourgeois, L. Silver segregation to θ' (Al₂Cu)–Al interfaces in Al–Cu–Ag alloys. *Acta Mater* **60**, 6033-6041 (2012).
2. Chen, B. A. et al. Effect of interfacial solute segregation on ductile fracture of Al–Cu–Sc alloys. *Acta Mater* **61**, 1676-1690 (2013).
3. Gao, Y. H. et al. Si-mediated reassembly of interfacially segregated Sc atoms in an Al–Cu–Sc alloy exposed to high-temperature creep. *J Alloy Compd* **845**, 156266 (2020).
4. Gao, Y. H. et al. Segregation-sandwiched stable interface suffocates nanoprecipitate coarsening to elevate creep resistance. *Mater Res Lett* **8**, 446-453 (2020).
5. Kumar Makineni, S. et al. Enhancing elevated temperature strength of copper containing aluminium alloys by forming L1(2) Al(3)Zr precipitates and nucleating theta" precipitates on them. *Sci Rep* **7**, 11154 (2017).
6. Gariboldi, E., Bassani, P., Albu, M.&Hofer, F. Presence of silver in the strengthening particles of an Al-Cu-Mg-Si-Zr-Ti-Ag alloy during severe overaging and creep. *Acta Mater* **125**, 50-57 (2017).
7. Ringer, S. P., Yeung, W., Muddle, B. C.&Polmear, I. J. Precipitate stability in Al-Cu-Mg-Ag alloys aged at high temperatures. *Acta Metall Mater* **42**, 1715-1725 (1994).
8. Hutchinson, C. R., Fan, X., Pennycook, S. J.&Shiflet, G. J. On the origin of the high coarsening resistance of Ω plates in Al–Cu–Mg–Ag Alloys. *Acta Mater* **49**, 2827-2841 (2001).
9. Nie, J. F.&Muddle, B. C. Strengthening of an Al–Cu–Sn alloy by deformation-resistant precipitate plates. *Acta Mater* **56**, 3490-3501 (2008).
10. Liu, L., Chen, J. H., Wang, S. B., Liu, C. H., Yang, S. S.&Wu, C. L. The effect of Si on precipitation in Al–Cu–Mg alloy with a high Cu/Mg ratio. *Mater Sci Eng A* **606**, 187-195 (2014).
11. Marioara, C. D. et al. Improving Thermal Stability in Cu-Containing Al-Mg-Si Alloys by Precipitate Optimization. *Metall Mater Trans A* **45**, 2938-2949 (2014).
12. Kang, S. J., Kim, Y.-W., Kim, M.&Zuo, J.-M. Determination of interfacial atomic structure, misfits and energetics of Ω phase in Al–Cu–Mg–Ag alloy. *Acta Mater* **81**, 501-511 (2014).
13. Kang, S. J., Zuo, J.-M., Han, H. N.&Kim, M. Ab initio study of growth mechanism of omega precipitates in Al-Cu-Mg-Ag alloy and similar systems. *J Alloy Compd* **737**, 207-212 (2018).
14. Koch, C. Determination of Core Structure Periodicity and Point Defect Density Along Dislocations. Arizona State University (2002).

15. Bouchet, J., Bottin, F., Jomard, G.&Zérah, G. Melting curve of aluminum up to 300 GPa obtained through ab initio molecular dynamics simulations. *Phys Rev B* **80**, (2009).
16. Hong, Q.-J.&van de Walle, A. Prediction of the material with highest known melting point from ab initio molecular dynamics calculations. *Phys Rev B* **92**, (2015).
17. Wenner, S., Jones, L., Marioara, C. D.&Holmestad, R. Atomic-resolution chemical mapping of ordered precipitates in Al alloys using energy-dispersive X-ray spectroscopy. *Micron* **96**, 103-111 (2017).

REVIEWER COMMENTS

Reviewer #1 (Remarks to the Author):

The authors have made a valiant attempt at addressing my comments. All my major concerns have been addressed satisfactorily. However I still have a few questions and suggestions:

1. The authors should use “partially covered” rather than “enveloped”, as the latter indicates near full coverage of the precipitates by other phases. This is not the case here.
2. Page 4 of Rebuttal: replace “it is believed” by “it is suggested”.
3. The authors didn’t address my question about how the concentrations were determined by EDX (see p11 of Rebuttal). Using an intensity tool in Velox for the raw data implies no background subtraction or peak fitting were performed. This is OK if only the presence of certain elements is to be determined, but not for determining the chemical composition.
4. In Fig. 3, I cannot see any Mg detected by EDX in the middle layer of the new interfacial phase (layer L2). This contradicts the crystal model presented in (j).

Reviewer #2 (Remarks to the Author):

I feel the authors have answered well and revised according to the comments given from the reviews -
So I suggest to publish this now -

One minor thing; page 20 line 7:

The atomic resolution HAADF images of alloy after thermal exposure were

Delete 'of alloy' or rephrase!

The rest - GO from me!

Response to reviewers' comments

Dear Editor and Reviewers,

The authors would like to thank you for the time and efforts that you spent on reviewing our manuscript. We gratefully appreciate for your valuable comments. All these comments are carefully considered and responded accordingly in the following section. Revisions are marked in **yellow background** in the manuscript and also in this response letter.

We sincerely hope these response and revisions can satisfy you.

Regards,

Yours sincerely,

Kai Li and Yong Du, on behalf of all authors

REVIEWER COMMENTS

Reviewer #1 (Remarks to the Author):

The authors have made a valiant attempt at addressing my comments. All my major concerns have been addressed satisfactorily. However I still have a few questions and suggestions:

1. The authors should use “partially covered” rather than “enveloped”, as the latter indicates near full coverage of the precipitates by other phases. This is not the case here.

Response: Thanks very much for your suggestion. The word “enveloped” has been replaced by “partially covered”. The revised parts are shown below:

“Here we obtain multiple interface structures in an Al-Cu-Mg-Ag-Si-Sc alloy including Sc segregation layers, C and L phases as well as a newly discovered χ -AgMg phase, **which partially cover** the θ' precipitates.” In line 16, page 1.

“This concept of **covering** precipitates with multiple interface phases and segregation layers provides an effective strategy for designing other heat resistant materials.” In line 21, page 1.

“the remaining Mg solutes combined with Ag solutes to form the newly discovered AgMg interface phase that **partially covers** the θ' precipitate.” In line 3, page 11.

“These calculations explain the high stability of θ' precipitates **partially covered** with Sc segregation layers observed in our work” in line 9, page 13.

In the second case, we tend not to add the word "partially" because we are looking forward to obtaining a microstructure with precipitates fully covered by these interface phases and segregation layers in the near future.

2. Page 4 of Rebuttal: replace “it is believed” by “it is suggested”.

Response: Thanks very much for your suggestion. It has been revised in the manuscript, which can be seen in line 13, page 15.

3. The authors didn't address my question about how the concentrations were determined by EDX (see p11 of Rebuttal). Using an intensity tool in Velox for the raw data implies no background subtraction or peak fitting were performed. This is OK if only the presence of certain elements is to be determined, but not for determining the chemical composition.

Figure. R1 The process of determining the concentration using EDX spectra. **a** Screenshot of Velox software manual. **b** Screenshot of parameters for determining the concentration of Mg in Velox software.

Response: Sorry for not explaining clearly in the previous Rebuttal. As shown in Fig. R1a, Velox software has four data views in the STEM-EDX mapping analysis module, namely *int*, *net*, *wt%* and *at%*. *Int* view shows the raw integrated spectra, while *net*, *wt%* and *at%* views show the quantified data which are background corrected and fitted. As shown in Fig. R1b, in our work, the concentration of Mg has been analyzed using the Intensity Profile tool under the *at%* view. The description of how concentration of Mg is determined has been updated in the *Methods* section of the manuscript. The revised part in lines 18-20, page 20 is shown below:

“... by applying the IntensityProfile tool in the Velox software to the quantified elemental-mapping data, which have been background corrected and peak fitted.”

4. In Fig. 3, I cannot see any Mg detected by EDX in the middle layer of the new interfacial phase (layer L2). This contradicts the crystal model presented in (j).

Response: Sorry for causing the inconvenience of reviewing this manuscript. In the last version of Fig. 3f, the Mg signal in L2 layer is weak and can hardly be noticed. One needs to go to very high magnifications to find the signal. Meanwhile, the concentration line profile of Mg shown in Fig. 3i, and the HAADF-STEM intensity line profiles shown in Figs. 3a and c, further confirm that the L2 layer contains Mg atoms. To display the signal of Mg more clearly, the brightness and contrast of Fig. 3f has been slightly adjusted in the current version. There are Mg signals at the positions marked by white dotted circles in L2 layer. We have rotated the model in Fig. 3j a bit in order to more straightforwardly compare it with Figs. 3a and 3e-h.

In addition, to ensure the accuracy and reproducibility of the experimental results, the atomic resolution EDX elemental maps from χ -AgMg interface phase on other precipitates are shown in Supplementary Fig. 4 (raw data are also uploaded). The Mg EDX maps shown in Supplementary Figs. 4b and e indicate that there are also Mg signals in these L2 layers.

All these information consistently demonstrate that the structure presented in Fig. 3j is accurate. The revised part in lines 1-3, page 8 is shown below:

As shown in Fig. 3f, Mg signals can be found at the atomic columns marked by white dotted circles in the L2 layer. Similar examples can be found in Supplementary Fig. 4.

Figure 3. **The structure of the newly discovered χ -AgMg interface phase.** **a-c** and **d** Atomic resolution HAADF-STEM images of the χ -AgMg phase along $[100]_{Al}$ and $[110]_{Al}$, respectively. **b** Enlargement image of (a). **e-h** Atomic resolution EDX elemental maps of the area shown in (a). The intensity line profiles of the middle layer (L2) of the χ -AgMg phase are inserted in (b) and (c). **i** Concentration line profile of Mg in the χ -AgMg phase along the white arrow, the integration width is 7 nm. **j** 3D model of the θ' /Al interface structure with the χ -AgMg phase. It should be noted the viewing direction is parallel to $[010]$ direction of the supercell in (j) for (a-b), $[100]$ for (c) and $[110]$ for (d). The HAADF-STEM images simulated by QSTEM using the constructed model along different directions are inserted in (a), (b) and (c). **The atomic columns marked with white dotted circles in (f) correspond to those similarly marked in (a).**

Supplementary Figure 4. **Atomic resolution EDX elemental maps of χ -AgMg interface phase on different θ' precipitates.** **a** Atomic resolution HAADF-STEM image of a θ' precipitate with χ -AgMg interface phase only on one side. **b** and **c** Atomic resolution EDX elemental maps of the area shown in (a). **d** Atomic resolution HAADF-STEM image of a θ' precipitate with χ -AgMg interface phase on both sides. **e-f** Atomic resolution EDX elemental maps of the area shown in (d). The atomic

columns marked with white circles in (b) and (e) correspond to those similarly marked in (a) and (d), respectively.

Reviewer #2 (Remarks to the Author):

I feel the authors have answered well and revised according to the comments given from the reviews - So I suggest to publish this now -

One minor thing; page 20 line 7:

**The atomic resolution HAADF images of alloy after thermal exposure were
Delete 'of alloy' or rephrase!**

Response: Thanks very much for your suggestion. The words “of alloy” has been deleted in the manuscript, as can be seen in line 7, page 20.

One additional modification

To better distinguish the newly discovered AgMg interface phase in this work from other AgMg segregation structures (e.g. the AgMg layers on Ω phase), we name the newly discovered AgMg interface phase χ , and have accordingly revised the manuscript.

REVIEWERS' COMMENTS

Reviewer #1 (Remarks to the Author):

The authors have addressed my comments satisfactorily.